# Elf1 promotes transcription-coupled repair in yeast by using its C-terminal domain to bind TFIIH

Kathiresan Selvam [1,7], Jun Xu [2,5,7], Hannah E. Wilson[1], Juntaek Oh [2,6], Qingrong Li [2], Dong Wang [2,3,4] ✉ & John J. Wyrick [1] ✉

Transcription coupled-nucleotide excision repair (TC-NER) removes DNA lesions that block RNA polymerase II (Pol II) transcription. A key step in TC-NER is the recruitment of the TFIIH complex, which initiates DNA unwinding and damage verification; however, the mechanism by which TFIIH is recruited during TC-NER, particularly in yeast, remains unclear. Here, we show that the C-terminal domain (CTD) of elongation factor-1 (Elf1) plays a critical role in TC-NER in yeast by binding TFIIH. Analysis of genome-wide repair of UV-induced cyclobutane pyrimidine dimers (CPDs) using CPD-seq indicates that the Elf1 CTD in yeast is required for efficient TC-NER. We show that the Elf1 CTD binds to the pleckstrin homology (PH) domain of the p62 subunit of TFIIH in vitro, and identify a putative TFIIH-interaction region (TIR) in the Elf1 CTD that is important for PH binding and TC-NER. The Elf1 TIR shows functional, structural, and sequence similarities to a conserved TIR in the mammalian UV sensitivity syndrome A (UVSSA) protein, which recruits TFIIH during TC-NER in mammalian cells. These findings suggest that the Elf1 CTD acts as a functional counterpart to mammalian UVSSA in TC-NER by recruiting TFIIH in response to Pol II stalling at DNA lesions.

Transcription by RNA polymerase II (Pol II) is essential for cellular viability but can be blocked by the presence of DNA lesions on the transcribed strand (TS) of genes. Transcription-blocking lesions (TBLs) include ultraviolet (UV) light-induced cyclobutane pyrimidine dimers (CPDs) and bulky, helix-distorting DNA adducts[1,2]. TBLs cause elongating Pol II to stall, triggering a cellular repair pathway known as transcription coupled-nucleotide excision repair (TC-NER)[3–6]. Efficient removal of TBLs by the TC-NER pathway is critical to genome stability and cellular homeostasis, as genetic defects in TC-NER result in elevated sensitivity to DNA damaging agents such as UV light and can cause development defects, neurodegeneration, and rapid aging

syndrome in humans[4,5,7]. However, despite its importance and nearly four decades of intense study, key gaps remain in our understanding of the basic mechanism of TC-NER in eukaryotic cells[2,8].

TC-NER is initiated when Pol II stalling at a DNA lesion is detected by the Cockayne Syndrome B (CSB) protein in human cells or its homolog Rad26 in yeast. CSB/Rad26 binding eventually leads to the critical step of recruiting TFIIH, a helicase and ATPase complex that is required for DNA unwinding and damage verification, and which promotes subsequent excision of the damaged DNA strand[1,2,9]. While CSB/Rad26 binds to stalled Pol II, neither directly binds to TFIIH. In human cells, CSB, in conjunction with Cockayne Syndrome A protein

[1]School of Molecular Biosciences, Washington State University, Pullman, WA, USA. [2]Division of Pharmaceutical Sciences, Skaggs School of Pharmacy and Pharmaceutical Sciences, University of California San Diego, La Jolla, CA, USA. [3]Department of Cellular & Molecular Medicine, University of California San Diego, La Jolla, CA, USA. [4]Department of Chemistry and Biochemistry, University of California San Diego, La Jolla, CA, USA. [5]Present address: Genetics and Metabolism Department, The Children's Hospital, School of Medicine, Zhejiang University, National Clinical Research Center for Child Health, Hangzhou, China. [6]Present address: Department of Pharmacy, College of Pharmacy, Kyung Hee University, Seoul, Republic of Korea. [7]These authors contributed equally: Kathiresan Selvam, Jun Xu. ✉e-mail: dongwang@ucsd.edu; jwyrick@wsu.edu

and its associated Cullin Ring Ubiquitin Ligase complex, promotes Pol II ubiquitination, as well as recruitment and ubiquitination of a third TC-NER factor known as UVSSA[10–12]. UVSSA then recruits TFIIH via its TFIIH interacting region (TIR), which directly binds to the pleckstrin homology (PH) domain of the p62 subunit of TFIIH[11,13]. Notably, the UVSSA TIR resembles TFIIH-interacting regions in XPC, which is required for global genomic-nucleotide excision repair (GG-NER) in human cells[9,14], and in transcription factors (e.g., TFIIEα, p53, and DP1) that recruit TFIIH during transcription initiation by also binding the PH domain of p62[13,15–18]. However, yeast and many other species lack a UVSSA homolog or other TIR-containing TC-NER factor(s). Hence, the mechanism by which TFIIH is recruited during TC-NER in these species is unknown. Moreover, previous studies have revealed that the TC-NER pathway in yeast continues to repair UV damage, albeit less efficiently, in the absence of the key TC-NER factor Rad26. However, the molecular mechanism by which Rad26-independent TC-NER is initiated in yeast remains unclear.

Recently, a new TC-NER factor known as ELOF1 in human cells and Elf1 in yeast was identified[19–21]. ELOF1/Elf1 are transcription elongation factors that are associated with Pol II during transcription elongation and stimulate elongation rates[19,20,22–26]. In yeast, Elf1 binds to the coding regions of genes in a transcription-dependent manner[24], and its genome-wide binding profile resembles that of other transcription elongation factors[25]. Consistently, structural studies indicate that Elf1 functions as a key component of the DNA entry channel in the yeast Pol II elongation complex[22,23]. In human cells, ELOF1 is required for TFIIH recruitment during TC-NER because it stimulates ubiquitination of stalled Pol II and subsequent recruitment and ubiquitination of

UVSSA[19,20]. While Elf1 also plays a critical role in TC-NER in the yeast *Saccharomyces cerevisiae*, the molecular mechanism(s) by which yeast Elf1 promotes TC-NER is unclear.

Here, we use both genome-wide repair studies in yeast (*S. cerevisiae*) and in vitro biochemistry to show that the C-terminal domain (CTD) of yeast Elf1 contains a TIR that binds the PH domain of the p62 subunit of TFIIH and promotes TC-NER in genes throughout the yeast genome.

## Results

### Elf1 is required for Rad26-independent TC-NER

Our previous study indicated that Elf1 not only plays a genome-wide role in TC-NER in WT cells but may also specifically contribute to Rad26-independent TC-NER, at least at the *RPB2* locus[19]. To characterize the role of Elf1 in Rad26-independent TC-NER across the yeast genome, we used the CPD-seq method[27,28] (Fig. 1a) to map the genome-wide distribution of CPD lesions both immediately after UV irradiation (0 h) and following 2 h of repair (2 h) in a *rad26Δ elf1Δ* double mutant. Analysis of putative lesion-forming dinucleotide sequences associated with CPD-seq reads revealed enrichment at CPD-forming dipyrimidines (i.e., TT followed by TC, CT, and CC) in the UV-exposed samples but not in the No UV control (Fig. 1b), consistent with our previous results[28,29]. To characterize the impact of *elf1Δ* mutant on Rad26-independent TC-NER, we analyzed the fraction of CPDs remaining following 2 h of repair (2 h) relative to 0 h control in both the *rad26Δ elf1Δ* double mutant and a previously published *rad26Δ* single mutant[30] (Fig. 1c, d). Analysis of the fraction of unrepaired CPDs adjacent to the transcription start site (TSS) of ~5200 yeast genes in a

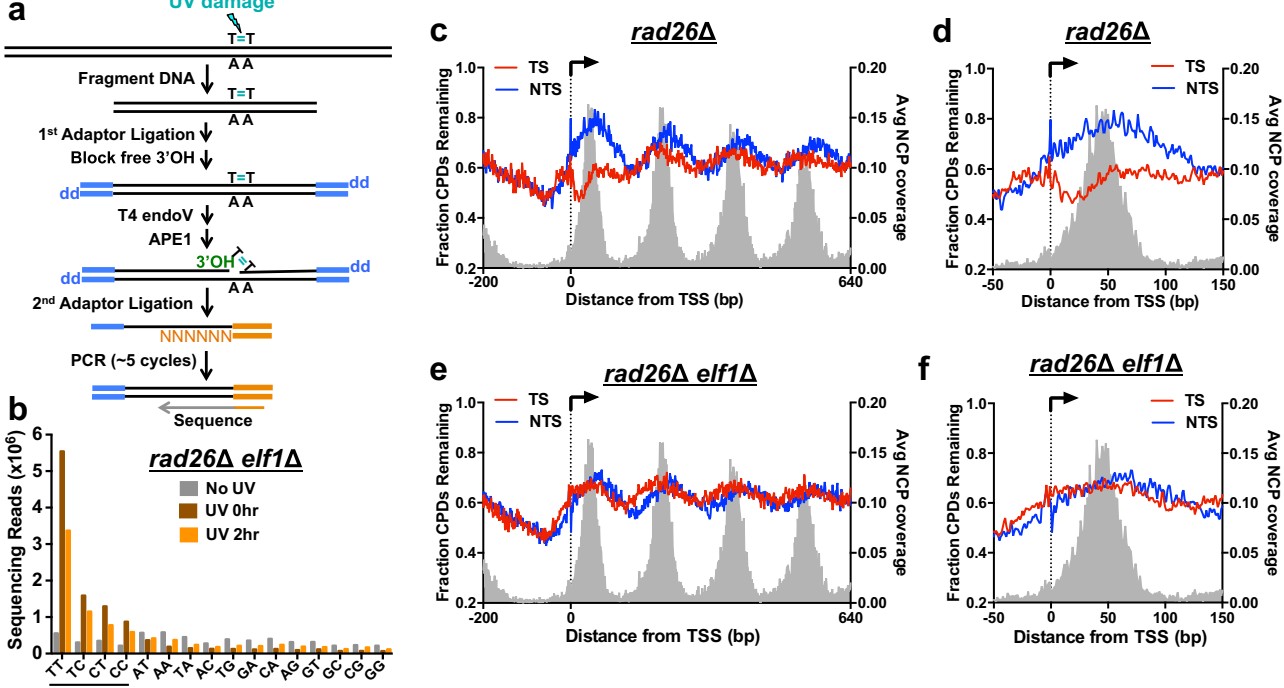

**Fig. 1 | Elf1 plays role in Rad26-independent TC-NER. a** Schematic of the CPD-seq protocol for mapping UV-induced CPD lesions at single nucleotide resolution. **b** CPD-seq reads in *elf1Δ rad26Δ* mutant cells are enriched at dipyrimidine sequences in UV-irradiated samples (0 h and 2 h) and not in non-irradiated (No UV) control. **c** Analysis of CPD-seq data from *rad26Δ* cells[30] near the transcription start site (TSS) for ~5200 genes at single nucleotide resolution on both the transcribed strand (TS) and non-transcribed strand (NTS). The data depicts regions 200 bp upstream of and 640 bp downstream from the TSS of each yeast gene. The number of unrepaired CPDs after 2 h of repair is normalized to the initial damage counts (0 h) to calculate the fraction of CPDs remaining. The gray peaks (right *y* axis) correspond to average nucleosome coverage from MNase nucleosome map[64]. NCP, nucleosome core particle. **d** CPD-seq data from *rad26Δ* (data from ref. 30) were analyzed immediately adjacent to the transcription start site (TSS) of ~5200 yeast genes on both the transcribed strand (TS) and non-transcribed strand (NTS). Fraction of unrepaired CPDs after 2 h repair relative to 0 h control is depicted. Gray background (right y-axis) depicts dyad positions of +1 nucleosome based on MNase nucleosome map[64]. **e** Same as (**c**), except analysis of CPD-seq data from *rad26Δ elf1Δ* cells. **f** Same as (**d**), except analysis of CPD-seq data from *elf1Δ rad26Δ*. Source data are provided as a Source Data file.

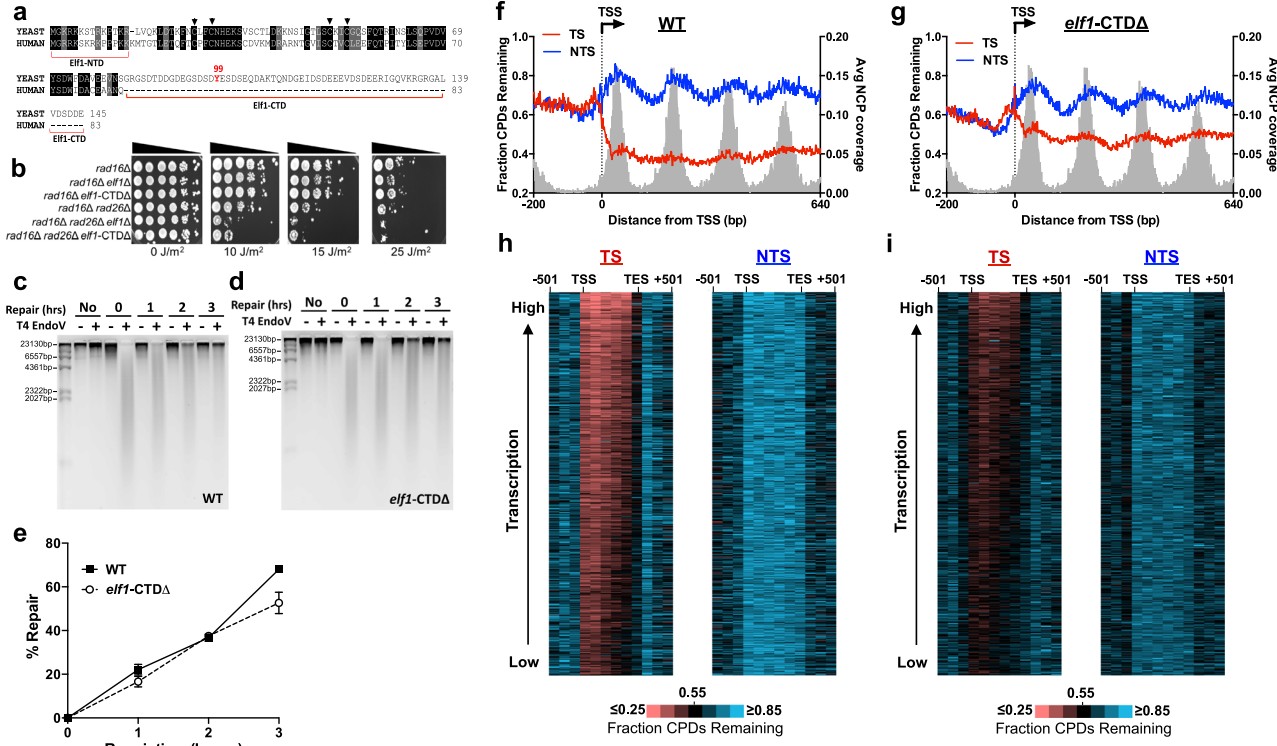

**Fig. 2 | Elf1-CTD is required for UV survival and efficient TC-NER in yeast.**
**a** Alignment of yeast Elf1 and human ELOF1 protein sequences. NTD, N-terminal domain; arrows, key cysteine residues in C4 zinc finger domain; and CTD, C-terminal domain. **b** Indicated mutant yeast strains were 10-fold serially diluted, spotted on the YPD plates, and exposed to the indicated doses of UVC light. Plates were photographed after 3 days of incubation in the dark. **c** Representative alkaline gel of bulk repair analysis of CPD lesions in WT cells at different time points following UVC irradiation. The genomic DNA was isolated at the indicated time points and treated with or without (+/−) T4 endonuclease V digestion and resolved on denaturing alkaline gels. First lane is the lambda DNA-HindIII digest molecular weight markers (Invitrogen). **d** Same as (**c**), except analysis of data from *elf1*-CTDΔ cells. **e** Quantification of the repair of CPDs in WT and *elf1*-CTDΔ yeast strains based on alkaline gel analysis. The plot represents mean ± SEM of three independent experiments. *$P \le 0.05$ analyzed by unpaired two-sided *t*-test with Holm–Sidak

correction. **f** Analysis of CPD-seq data from WT cells[19] for ~5200 genes at single nucleotide resolution on both the transcribed strand (TS) and non-transcribed strand (NTS). The plot depicts regions, 200 bp upstream of and +640 bp downstream from the transcription start site (TSS). The number of unrepaired CPDs after 2 h of repair is divided by the initial damage counts (0 h) to depict the fraction of CPDs remaining. The fraction of CPDs remaining was normalized to the bulk fraction of CPDs remaining determined by alkaline gel analysis. **g** Same as (**f**), except analysis of data from *elf1*-CTDΔ cells. **h**, **i** Gene cluster plot analysis of CPD-seq repair data from **h** WT and **i** *elf1*-CTDΔ cells for ~4500 genes, ordered based on transcription frequency[69]. Plots depict the fraction of unrepaired CPDs after 2 h of repair relative to the initial damage counts (0 h) for both TS and NTS. Data was normalized using alkaline gel analysis data, as described above. Source data are provided as a Source Data file.

*rad26*Δ mutant[30] revealed a defect in repair of the TS, with peaks of unrepaired CPDs along both the TS and non-transcribed strand (NTS) associated with the centers of yeast nucleosomes (Fig. 1c), likely due to nucleosome inhibition of CPD repair by the GG-NER pathway[28,29,31–33]. However, there were still fewer unrepaired CPDs along the TS than the NTS in the *rad26*Δ strain immediately downstream of the TSS (-120 base pairs (bp)) and particularly on the TSS-proximal side of the +1 nucleosome (Fig. 1c, d and Supplementary Fig. 1a). Analysis of published CPD-seq data for catalytically inactive ATPase mutant in rad26 (i.e., *rad26*-K328R), which is also defective in TC-NER[30], showed similar results (Supplementary Fig. 1b). These findings are consistent with previous analysis[30], and are indicative of Rad26-independent TC-NER near the TSS of yeast genes. In contrast, CPD-seq data for *rad26*Δ *elf1*Δ double mutant revealed a higher fraction of CPDs remaining on the TS immediately downstream of the TSS, so that the fraction of unrepaired CPDs along the TS was similar to the NTS (Fig. 1e, f). Taken together, these findings indicate that Elf1 is required for Rad26-independent TC-NER across the yeast genome.

**Deletion of the Elf1 C-terminal domain imparts UV sensitivity in GG-NER deficient yeast**
Inspection of the yeast Elf1 protein sequence revealed not only N-terminal and C4 zinc finger core domains, which are also present

in human ELOF1, but also an extended CTD that is absent from the human ELOF1 sequence (Fig. 2a). To characterize the role of these different domains in TC-NER, we mutated each domain in a GG-NER deficient *rad16*Δ mutant background since it has been previously shown that deletion of TC-NER factors typically imparts significant UV sensitivity only in cells lacking GG-NER[34,35]. UV spotting assays revealed that while deletion of the Elf1 N-terminus (*elf1*-NΔ, corresponding to deletion of residues 2–16) did not affect UV sensitivity in the *rad16*Δ background, mutating critical cysteine residues in the Elf1 zinc finger domain (*elf1*-ZnΔ, corresponding to *elf1*-C49A, C52A) or deletion of the Elf1 CTD (*elf1*-CTDΔ, corresponding to deletion of residues 85–145) significantly enhanced UV sensitivity (Supplementary Fig. 2a and Fig. 2b). To determine if these Elf1 domains also affected Rad26-independent TC-NER, we analyzed their UV sensitivity in a *rad16*Δ *rad26*Δ background. Our results indicated that both the *elf1*-ZnΔ and *elf1*-CTDΔ cells exhibited increased UV sensitivity in *rad16*Δ *rad26*Δ strains (Supplementary Fig. 2b and Fig. 2b), indicating these domains of Elf1 are important for Rad26-independent TC-NER.

While the zinc finger domain of Elf1 (and ELOF1) is required for binding to Pol II[19,20,22,23,36], the function of the Elf1 CTD is unknown. To test whether the Elf1-CTD is required for Elf1 protein stability, we performed western blot analysis of yeast extracts derived from strains

with tagged Elf1. These data indicated that the *elf1-CTDΔ* mutant did not affect Elf1 protein levels (Supplementary Fig. 3a), suggesting that the Elf1 CTD may instead play a direct functional role in TC-NER. Deletion of *ELF1* is lethal in combination with mutations in genes encoding other transcription elongation factors, such as Spt4[24], consistent with a role for Elf1 in transcription elongation. While our data confirmed that the *elf1Δ spt4Δ* double mutant is lethal in yeast, as it is unable to grow on media containing 5-FOA, the *elf1-CTDΔ spt4Δ* double mutant grew normally in medium with 5-FOA (Supplementary Fig. 3b). This observation suggests that, unlike full-length Elf1, the Elf1 CTD is not required for transcription elongation, and that the UV sensitivity of the *elf1-CTDΔ* mutant does not arise from a defect in transcription elongation.

## Elf1 C-terminal domain is required for efficient TC-NER

Since the UV sensitivity assays indicated the Elf1 CTD may play a role in TC-NER, we analyzed the impact of the Elf1 CTD on repair of UV-induced CPD lesions. Our initial assay used a T4 endonuclease V (T4 endoV) digestion and alkaline gel electrophoresis assay to measure repair of CPDs in bulk genomic DNA (Fig. 2c, d). The results indicated that the *elf1-CTDΔ* mutant did not cause a significant repair defect in bulk genomic DNA relative to WT ($P > 0.05$, Fig. 2e), suggesting that the Elf1-CTD is likely not required for global repair of UV lesions by the GG-NER pathway, which is primarily responsible for repair of CPD lesions in yeast.

To specifically study the role of Elf1-CTD in TC-NER, we analyzed the repair of CPDs across the genome in *elf1-CTDΔ* using CPD-seq. We examined repair of CPDs at single-nucleotide resolution around the TSS of ~5200 yeast genes for both the *elf1-CTDΔ* cells and a previously published WT control[19], which were normalized using the T4 endoV alkaline gel data (Fig. 2e). In WT cells, there were fewer unrepaired CPDs remaining after 2 h repair on the TS of yeast genes than the NTS, and unrepaired CPDs on the NTS were elevated near the central dyad of nucleosomes (Fig. 2f). In the *elf1-CTDΔ* mutant, the fraction of unrepaired CPDs along the TS was elevated relative to WT, particularly near the TSS of yeast genes, while repair of the NTS was largely unaffected (Fig. 2g). These data reveal that, similar to our published *elf1Δ* data[19], deletion of the Elf1-CTD causes a defect in the repair of the TS of yeast genes.

To further characterize the role of the Elf1-CTD on TC-NER, we analyzed the fraction of unrepaired CPDs on both the TS and NTS in six equally sized bins between the TSS and transcription end site (TES) of ~5000 yeast genes. In the *elf1-CTDΔ* strain, the fraction of CPDs remaining on the NTS after 2 h repair was roughly similar to WT (Supplementary Fig. 4a, b). However, there were more unrepaired CPDs along the TS in the *elf1-CTDΔ* strain throughout the transcribed region (i.e., TSS to TES). To test whether the Elf1 CTD is required for repair of the TS of all yeast genes or just a subset, we performed gene plot analysis of the fraction of the CPDs remaining after 2 h repair across each of the ~5000 yeast genes. The results revealed significantly more unrepaired CPDs along the TS in the *elf1-CTDΔ* mutant relative to WT for nearly all yeast genes (Fig. 2h, i). Taken together, these findings indicate that the Elf1 CTD is required for efficient repair of the TS of nearly all yeast genes.

CPDs located on the TS of yeast genes are repaired by both the TC-NER and GG-NER pathways. To specifically measure the impact of the Elf1 CTD on TC-NER, we analyzed asymmetry in repair of the transcribed and non-transcribed DNA strands, which quantifies TC-NER activity, as TC-NER only repairs the TS. Repair asymmetry can be quantified using the $\log_2$ ratio of unrepaired CPDs on the TS relative to the NTS (see "Methods"), as described previously[29]. In the *elf1-CTDΔ* mutant, the average $\log_2$ TS/NTS ratio for the six transcribed bins was −0.34, lower than the −0.67 value for WT (Supplementary Fig. 4c), indicating reduced repair asymmetry. This analysis is consistent with the hypothesis that the Elf1 CTD regulates TC-NER.

We also used CPD-seq to measure the impact of the *elf1-CTDΔ* mutant on repair in a *rad16Δ* background, which is deficient in GG-NER. In the *rad16Δ elf1-CTDΔ* double mutant, there were more unrepaired CPDs along the NTS of ~5200 yeast genes after 2 h repair than the *elf1-CTDΔ* mutant alone, and the absence of the periodic pattern of unrepaired CPDs associated with nucleosomes (Fig. 3a), consistent with a loss of GG-NER activity[33]. The *rad16Δ elf1-CTDΔ* strain showed a marginal defect in the repair of the TS (Supplementary Fig. 5a), as quantified by an average $\log_2$ TS/NTS ratio of the transcribed bins of −0.68 for the *rad16Δ elf1-CTDΔ* mutant, relative to average $\log_2$ TS/NTS ratios of −0.86 and −1.04 for *rad16Δ* controls[32,33] (Fig. 3b and Supplementary Fig. 5b), indicating diminished repair asymmetry.

Our UV sensitivity data indicate that the Elf1 CTD may also be required for Rad26-independent TC-NER (Fig. 2b). To test this hypothesis, we analyzed repair in *rad26Δ elf1-CTDΔ* using CPD-seq and compared the repair to our published *rad26Δ* CPD-seq data[30]. Analysis of the CPD-seq data near the TSS of ~5200 yeast genes revealed a higher fraction of CPDs remaining along the TS in *rad26Δ elf1-CTDΔ* cells (Fig. 3c) compared to *rad26Δ* cells[30], particularly immediately downstream of the TSS (compare Fig. 3d with Fig. 1d). The role of the Elf1 CTD in Rad26-independent TC-NER was further confirmed by diminished asymmetry of repair in transcribed bins in the *rad26Δ elf1-CTDΔ* compared to *rad26Δ* alone (Supplementary Fig. 5c and Fig. 3e), as reflected in the average $\log_2$ TS/NTS ratio of 0.069 for the *rad26Δ elf1-CTDΔ* mutant, but −0.061 for the *rad26Δ* control. A similar loss of repair asymmetry was observed relative to the catalytically inactive rad26-K328R mutant (Supplementary Fig. 5d, average $\log_2$ TS/NTS ratio of −0.096). However, the *elf1-CTDΔ* (or *elf1Δ*) in a *rad16Δ rad26Δ* background was less UV sensitive than an *rad16Δ rad26Δ rpb9Δ* mutant (Supplementary Fig. 6), a mutant that should render cells completely defective in TC-NER[37]. These findings indicate that the *elf1-CTDΔ* mutant significantly reduces but likely does not completely eliminate Rad26-independent TC-NER, consistent with a previous analysis of *elf1Δ* mutant cells[19]. In summary, these results imply that the Elf1 CTD plays an important role in TC-NER in WT and *rad26Δ* mutant cells across the yeast genome.

## Elf1 CTD binds to the PH domain of the p62 subunit of TFIIH

Previous studies indicate that human ELOF1 facilitates TFIIH recruitment indirectly by promoting Pol II ubiquitination and UVSSA recruitment[19,20]. Since yeast lacks a UVSSA homolog, we wondered if Elf1 might directly interact with TFIIH, perhaps through its CTD. To test this hypothesis, we expressed and purified either full-length Elf1, Elf1 without CTD, or Elf1 CTD alone fused to glutathione S-transferase (GST) and measured their binding to purified PH domain of the p62 subunit of TFIIH in vitro. We tested binding to the PH domain of the p62 subunit since previous studies have shown that both transcription factors (e.g., TFIIE and p53) and NER factors, including Rad4/XPC and UVSSA, specifically bind to the PH domain to recruit TFIIH[13,15–18]. SDS-PAGE analysis of GST pull-down assays revealed that while full-length or the N-terminal half of Elf1 (i.e., residues 1–85) did not bind the PH domain of p62 in vitro, the Elf1 CTD (residues 85–145) did bind the PH domain (Fig. 4a). That full-length Elf1 was unable to bind the PH domain suggests that the Elf1 CTD may be inhibited by other regions of Elf1, which may have important ramifications for Elf1 activity during TC-NER (see "Discussion"). Quantitative binding assays of GST-Elf1-CTD and purified PH domain using Biolayer Interferometry (BLI) yielded a dissociation constant ($K_d$) of $3.82 \pm 1.09\ \mu M$ (mean ± standard deviation; see Supplementary Fig. 7 and Supplementary Table 1).

To determine which region(s) of the Elf1 CTD are important for TFIIH binding, we expressed and purified different truncated versions of the Elf1 CTD fused to GST, and tested their binding to the PH domain of p62 in vitro. The GST pull-down data (Fig. 4b,c) indicated that Elf1 residues 85–122 and 85–113 bound to the PH domain similarly as the

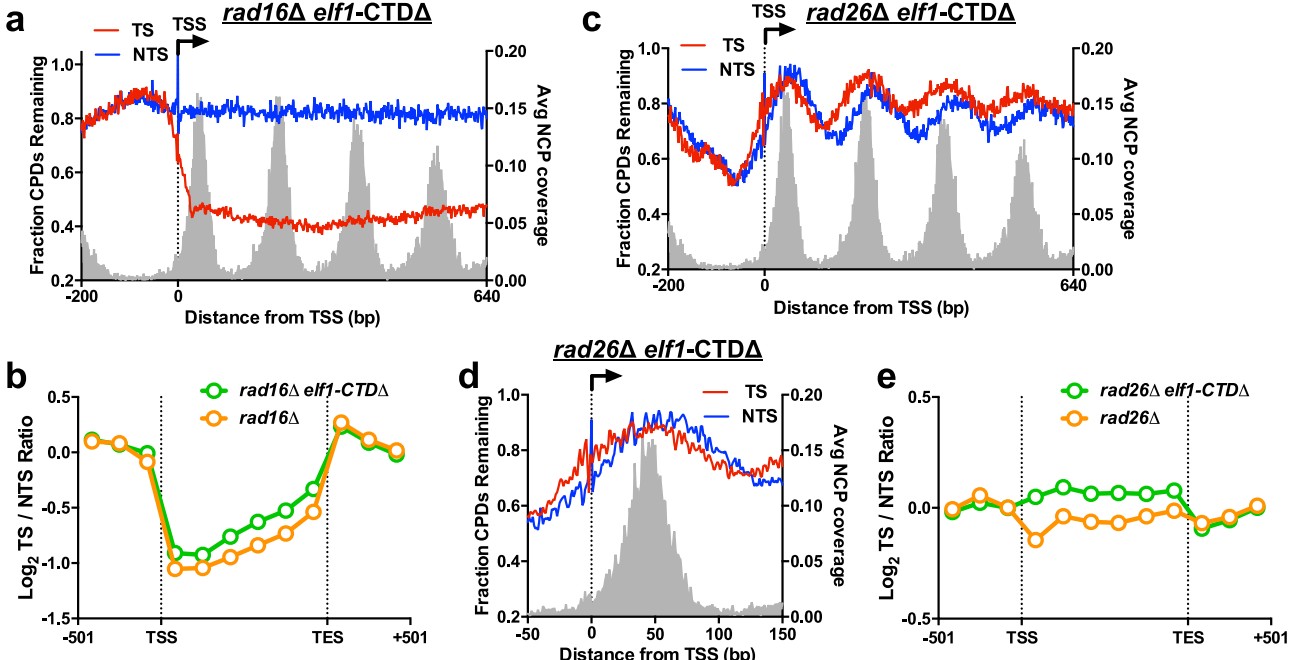

**Fig. 3 | Elf1-CTD promotes TC-NER across the genome. a** Analysis of CPD-seq data from *rad16Δ elf1*-CTDΔ cells adjacent to the TSS for ~5200 genes. The fraction of unrepaired CPDs after 2 h of repair relative to the 0 h control is plotted. Nucleosome positioning data from ref. 64. **b** Log₂ ratio of unrepaired CPDs (after 2 h repair relative to 0 h control) on the TS relative to the NTS in the *rad16Δ elf1*-CTDΔ and *rad16Δ* cells plotted between TSS and TES of ~5000 genes. Data for *rad16Δ* cells from ref. 33. Each gene was divided in six equally sized bins, and three 167 bp bins in flanking DNA upstream of the TSS and downstream of the transcription end site (TES) are also depicted. **c** Same as (**a**), except CPD-seq data from *rad26Δ elf1*-CTDΔ cells are depicted. **d** Close up of CPD-seq data from *rad26Δ elf1*-CTDΔ cells immediately flanking the TSS and +1 nucleosome of ~5200 yeast genes. **e** Same as (**b**), except analysis of data from *rad26Δ elf1*-CTDΔ and *rad26Δ* cells. Data for *rad26Δ* cells from ref. 30. Source data are provided as a Source Data file.

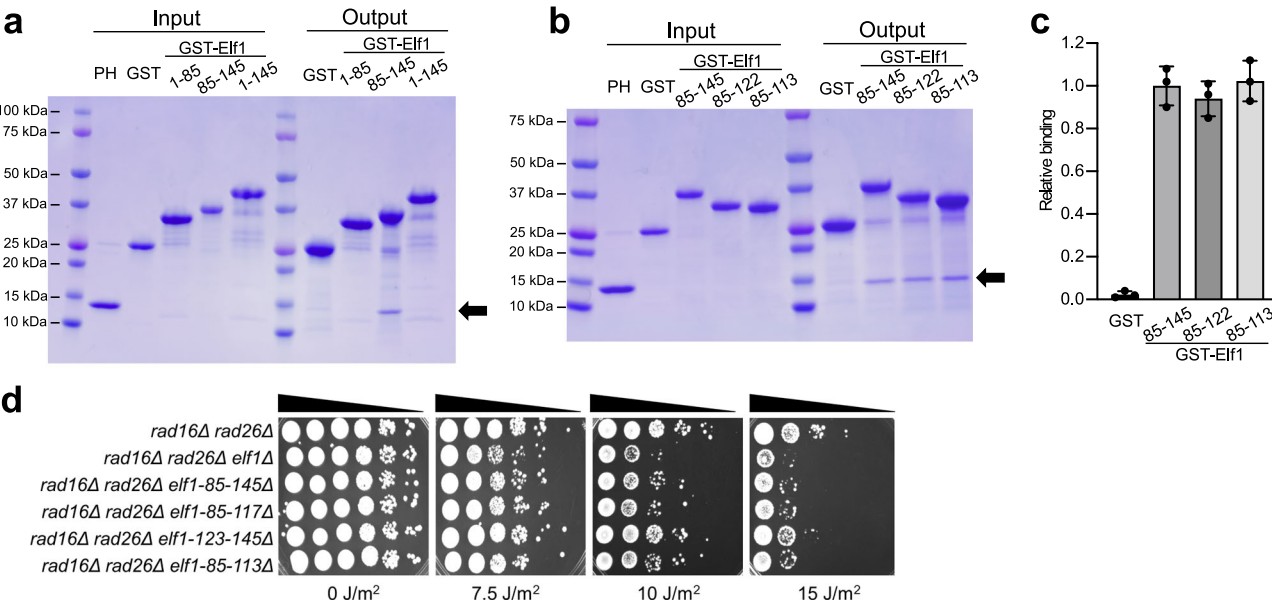

**Fig. 4 | Elf1 CTD interacts with the PH domain of p62 subunit of TFIIH. a, b** SDS polyacrylamide gel electrophoresis (PAGE) analysis of pull-down assays, in which purified Glutathione S-transferase (GST) fusion proteins containing the indicated Elf1 regions were incubated with purified pleckstrin homology (PH) domain of the p62 subunit of TFIIH (see Input). Output indicates proteins present after GST pull-down. Band corresponding to PH domain is indicated with arrow. Only the Elf1-CTD interacts with the PH domain. Lanes 1 and 7 in (**a, b**) are Precision Plus Protein Dual Color Standards molecular weight (MW) markers (Bio-Rad). The experiment was repeated three times with similar results. **c** Quantification of pull-down assay based on SDS-PAGE analysis in (**b**). Data depicted in (**c**) are mean and standard deviation (*n* = 3). **d** UV spotting assay of *elf1* mutant strains exposed to the indicated doses of UV light. Source data are provided as a Source Data file.

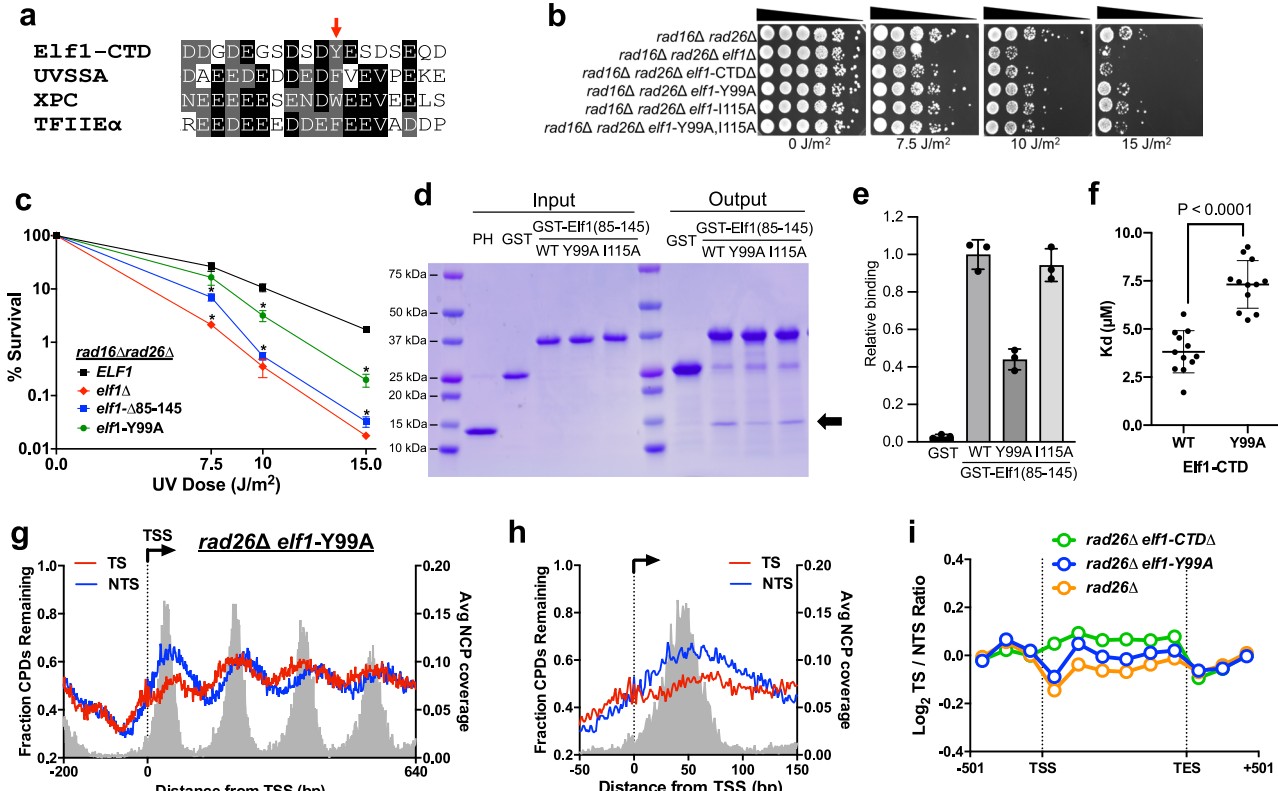

**Fig. 5 | Conserved TIR within the Elf1 CTD is important for PH domain binding and TC-NER. a** Alignment of putative TFIIH-interacting region (TIR) in yeast Elf1-CTD (residues 89–106) with TIRs from other PH domain interacting proteins, namely UVSSA (residues 398–415), XPC (residues 123–140), TFIIEα (residues 377–394). Red arrow indicates location of conserved aromatic residue critical for PH binding. **b** UV spotting assay of *elf1*-CTD mutant strains exposed to the indicated doses of UV light. **c** Quantitative UV survival assay of the indicated *elf1* mutants (or *ELF1* WT) in a *rad16 rad26Δ* background. The graph represents the quantification of UV survival from three (*n* = 3) independent experiments. Mean ± SEM is depicted, *P ≤ 0.05, based on two-sided *t*-test with Holm–Sidak correction for different UV doses for each pair of strains compared. **d** SDS-PAGE analysis of GST pull-down assays in which the purified GST-Elf1-CTD (residues 85–145) fusion proteins containing the indicated WT or mutant sequences were incubated with purified PH domain of p62 subunit of TFIIH (indicated by the arrow). Lanes 1 and 7 are Precision Plus Protein Dual Color Standards molecular weight (MW) markers (Bio-Rad). **e** Quantification of Elf1-CTD interaction with PH domain from replicate GST pull-down experiments. Mean and standard deviation (*n* = 3) are depicted. **f** Biolayer Interferometry (BLI) assay was used to determine the dissociation constant ($K_d$) values for the interaction between PH domain of p62 and WT and Y99A GST-Elf1-CTD (residues 85–145). $K_d$ measurements are for three independent assays conducted at four concentrations of PH-p62, ranging from 30 μM to 3.75 μM, for a total of twelve data points per variant. *P*-value is $2.49 \times 10^{-7}$ from two-sided *t*-test. Error bars represent standard deviation. **g** Analysis of CPD-seq data after 2 h repair relative 0 h control in *rad26Δ elf1*-Y99A cells adjacent to the TSS of ~5200 genes. Average nucleosome coverage from ref. 64. **h** Close up of CPD-seq data from *rad26Δ elf1*-Y99A cells immediately flanking the TSS of ~5200 yeast genes. **i** Plot of transcriptional asymmetry in repair of TS and NTS of ~5000 genes using CPD-seq data from *rad26Δ*[30], *rad26Δ elf1*-Y99A, and *rad26Δ elf1*-CTDΔ cells.

Elf1-CTD (i.e., residues 85–145), indicating that Elf1 residues 85–113 likely contain the TFIIH-interacting region. Deletion of Elf1 residues 85–113 in yeast imparted elevated UV sensitivity in a *rad16 rad26Δ* mutant background (Fig. 4d), suggesting that this region is important for TC-NER in yeast.

### Elf1 CTD has a TFIIH-interacting region important for Rad26-independent TC-NER

Multiple sequence alignment of the Elf1 sequences from related yeast species indicated that residues 95–106 in the CTD are highly conserved (Supplementary Fig. 8). This region contains a conserved aromatic residue (Y99) flanked by acidic residues. This region, as well as the rest of the Elf1 CTD, is predicted by the PONDR software[38] to be intrinsically disordered (Supplementary Fig. 9). Notably, these sequence features are characteristic of TFIIH-interacting regions (TIRs) found in XPC, UVSSA, and other proteins that bind the PH domain of p62 subunit of TFIIH[13,15–18]. Sequence alignment of the putative Elf1 TIR (residues 89–106) with TIRs from XPC, UVSSA, and other TFIIH-interacting proteins showed considerable similarity (Fig. 5a), with each TIR containing a central aromatic residue (Y99 in Elf1) flanked by acidic residues.

Since previous studies have suggested that the central aromatic residue in the TIR is important for binding TFIIH[13,16–18], we used CRISPR genome editing[39,40] to construct an *elf1*-Y99A mutant and tested its UV sensitivity in *rad16* and *rad16 rad26Δ* mutant backgrounds. Spotting assays indicated that the *elf1*-Y99A caused a mild-to-moderate increase in UV sensitivity in these mutant backgrounds (Fig. 5b and Supplementary Fig. 10). Quantitative analysis of the UV sensitivity of the *elf1*-Y99A in a *rad16 rad26Δ* mutant background revealed that the *elf1*-Y99A mutant imparted significant UV sensitivity, although to a lesser extent than the *elf1Δ* or *elf1*-CTDΔ (85-145Δ) mutants (Fig. 5c), suggesting other residues in the Elf1 CTD also play a role in UV resistance. In summary, these UV sensitivity assays indicate that Elf1-Y99 residue in the putative TIR plays a role in TC-NER.

To determine whether the putative Elf1 TIR mediates binding to TFIIH, we expressed and purified a GST-fused Elf1-CTD containing the Y99A mutant and tested its binding to the PH domain of p62. As a control, we also expressed and purified a GST-Elf1-CTD containing a mutation in a hydrophobic residue that lies outside the putative TIR (i.e., Elf1-I115, see Supplementary Fig. 8). GST pull-down assays indicated that the Y99A mutant caused a significant decrease in binding to the p62 PH domain relative to the WT CTD (Fig. 5d, e). In contrast, the

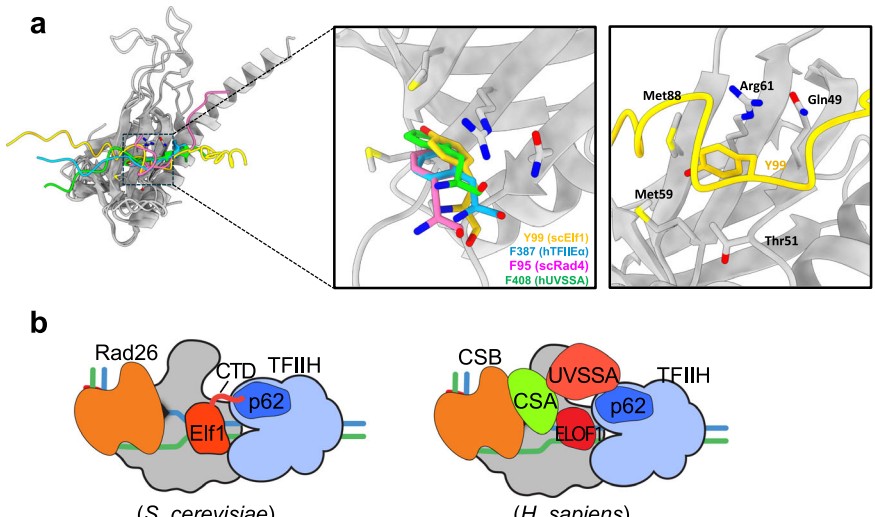

**Fig. 6 | Model of Elf1 CTD bound to p62 subunit of TFIIH. a** Superposition of Elf1-CTD (yellow), Rad4 (magenta), TFIIEα (blue), and UVSSA (green) in binding the PH domain of p62 (grey). The model of *Saccharomyces cerevisiae* Elf1-CTD/p62 was generated by Alphafold-multimer[42]. The NMR models of *Saccharomyces cerevisiae* Rad4/p62 (PDB: 2M14), *Homo sapiens* TFIIEα/p62 (PDB: 2RNR), and *Homo sapiens* UVSSA (PDB: 5XV8) were used in the structural alignment. In the left panel with zoom-in views, the key residues, Y99 in Elf1-CTD, F95 in Rad4, F387 in TFIIEα, and F408 in UVSSA, are depicted as sticks. The p62 residues involved in the interaction with Elf1-CTD are also visualized as sticks. **b** Comparison of TFIIH recruitment between human and yeast during TC-NER. We propose the Elf1-CTD plays an equivalent role as human UVSSA to recruit TFIIH.

I115A control did not significantly affect binding (Fig. 5d, e). Quantitative BLI assays indicated that the $K_d$ of the GST-Elf1-CTD Y99A mutant increased to $7.32 \pm 1.24\,\mu M$ (Fig. 5f and Supplementary Table 2), confirming that the Y99A mutant CTD has a significantly weaker interaction with the p62 PH domain than the WT CTD ($P < 0.0001$, Fig. 5f). Taken together, these findings indicate that the Y99 residue in the putative Elf1 TIR is important for binding to TFIIH.

To test whether the *elf1*-Y99A mutant affected TC-NER, we used CPD-seq to measure repair of UV damage. We analyzed repair in a *rad26Δ* mutant background since the *rad16Δ rad26Δ elf1*-Y99A mutant showed elevated UV sensitivity (Fig. 5b, c). Analysis of the CPD-seq data revealed a partial defect in repair of the TS relative to the NTS, particularly immediately downstream of the TSS (Fig. 5g, h). Moreover, the transcriptional asymmetry along the transcribed regions of ~5000 yeast genes also showed a partial defect in the *rad26Δ elf1*-Y99A double mutant relative to the *rad26Δ* or *rad26*-K328R mutants alone (Fig. 5i and Supplementary Fig. 11), as reflected in an average $\log_2$ TS/NTS of −0.0049 (compared to −0.061 and −0.096 for *rad26Δ* and *rad26*-K328R). However, the defect in transcriptional asymmetry was not as significant as observed in *rad26Δ elf1*-CTDΔ double mutant (average $\log_2$ TS/NTS of 0.069, see also Fig. 5i), consistent with our UV sensitivity results (Fig. 5c) and in vitro binding assays (Fig. 5d, e).

To determine whether Elf1 Y99 promotes UV resistance through its interaction with the PH domain of the p62 subunit of TFIIH (encoded by the yeast *TFB1* gene), CRISPR was used to specifically delete the PH domain (residues 2–114) in *TFB1*. UV spotting assays indicated that the *tfb1*-PHΔ mutant showed elevated UV sensitivity in a *rad16Δ* mutant background (Supplementary Fig. 12a), suggesting that the PH domain plays a role in TC-NER. The *tfb1*-PHΔ mutant also showed elevated UV sensitivity in a *rad16Δ rad26Δ* mutant background (Supplementary Fig. 12b), indicating that the PH domain also functions in Rad26-independent TC-NER. These findings are consistent with a recent report indicating that the Tfb1-PH domain regulates TC-NER in yeast[41]. Mutating *elf1*-Y99A did not increase the sensitivity of the *tfb1*-PHΔ mutant in either a *rad16Δ* or *rad16Δ rad26Δ* mutant background. (Supplementary Fig. 12), suggesting that the *elf1*-Y99A and *tfb1*-PHΔ mutants in these genetic backgrounds are epistatic. To confirm this finding, we performed quantitative UV sensitivity assays on each

mutant strain in a *rad16Δ rad26Δ* mutant background. The *elf1*-Y99A mutant caused a significant increase in UV sensitivity at the 10 J/m² dose ($P < 0.05$) and a marginal increase in UV sensitivity at the 15 J/m² dose. In contrast, the *elf1*-Y99A mutant did not significantly alter the UV sensitivity of the *tfb1*-PHΔ mutant in a *rad16Δ rad26Δ* background (Supplementary Fig. 12c), confirming these two mutants are epistatic. The *elf1*-CTDΔ mutant also appeared to be largely epistatic with the *tfb1*-PHΔ mutant (Supplementary Fig. 12a, b). Quantitative UV sensitivity assays indicated that the *elf1*-CTDΔ caused a significant increase in the UV sensitivity of the *tfb1*-PHΔ mutant in a *rad16Δ rad26Δ* background at the 7.5 and 10 J/m² UV doses ($P < 0.05$, Supplementary Fig. 12d). However, the magnitude of the increase in sensitivity across all UV doses (~2.6-fold) was much less than observed in cells in which the TFB1-PH domain was intact (~23-fold; see Supplementary Fig. 12d), suggesting that the Elf1-CTD is largely epistatic with the Tfb1-PH domain in a *rad16Δ rad26Δ* background. In summary, these findings support a model in which the Y99 residue in the Elf1 CTD promotes UV resistance by interacting with the PH domain in TFIIH to facilitate TC-NER.

Finally, AlphaFold-multimer[42] was used to predict the structure of the binding interaction between the Elf1-CTD and the PH domain of p62 (Fig. 6a). The resulting structure suggests that the Elf1-CTD binds to the PH domain primarily through Tyr99, which inserts in a pocket on the PH domain's surface composed of β-sheets. This pocket, characterized by a hydrophobic environment, appears to stabilize the interaction. Notably, Tyr99 is observed to stack with Arg61 of the PH domain at an approximate distance of 3.3 Å. This structural prediction is consistent with our biochemical and genetic data. Importantly, alignment of the predicted Elf1-CTD/p62 interface with NMR structures of known TFIIH-interacting proteins bound to the PH domain of p62 (i.e., human UVSSA-TIR/p62 (PDB: 5XV8), yeast Rad4-TIR/p62 (PDB: 2M14), and human TFIIEα-TIR/p62 (PDB: 2RNR)) reveals a striking common binding mode of how the TIR binds to p62. Specifically, conserved aromatic residues (Tyr99 in Elf1, Phe408 in UVSSA, Phe95 in Rad4, and Phe387 in TFIIEα) bind to the same pocket of p62.

Taken together, these findings suggest that the TIR of Elf1-CTD functions in a similar manner as other known TFIIH-interacting proteins (such as UVSSA, Rad4/XPC, and TFIIEα) in the recruitment of

TFIIH. This is consistent with our biochemical data indicating that the Elf1 CTD binds to the PH domain of the TFIIH p62 subunit, and our CPD-seq data indicating that the Elf1-CTD is required for efficient repair of the TSs of yeast genes, both in *RAD26* (WT) and *rad26Δ* cells.

## Discussion

TC-NER plays a critical role in maintaining genome stability by removing DNA lesions that would otherwise impede Pol II transcription. We and others have previously identified the elongation factor Elf1 as playing an important role in TC-NER in yeast[19,43], but the molecular mechanism by which Elf1 promotes TC-NER was previously unclear. Here, we have shown that the intrinsically disordered CTD of Elf1 promotes TC-NER across the yeast genome. Our data indicate that the Elf1 CTD binds the PH domain of the p62 subunit of TFIIH in vitro, suggesting it may play a direct role in recruiting TFIIH to initiate repair. We further demonstrate that the Elf1 CTD sequence contains a TFIIH-interacting region (TIR) that resembles similar regions in the human NER factors XPC and UVSSA and that the Elf1 TIR is important for TFIIH binding and TC-NER in yeast. Taken together, these findings suggest a model in which Elf1 recruits the TFIIH complex during TC-NER by directly binding the p62 PH domain (Fig. 6b).

Recruitment of the TFIIH complex is a critical step during NER since TFIIH both unwinds the DNA in preparation for incision of the damaged strand and verifies that damage is present[2,6,9]. During GG-NER, TFIIH is recruited by the key damage sensor proteins XPC in human cells and Rad4 in yeast[9,14]. However, the protein responsible for recruiting TFIIH during TC-NER in yeast was previously unclear. Our data indicate that the Elf1 CTD likely promotes TFIIH recruitment during repair by directly binding the PH domain of the p62 subunit of TFIIH. This same PH domain is bound by XPC/Rad4 to promote TFIIH recruitment during GG-NER[17,44], suggesting that a common biochemical mechanism is utilized to recruit TFIIH in each NER subpathway. Since Elf1 proteins in many other species, including yeasts (e.g., *S. pombe*) and plants (e.g., *Arabidopsis*), have an extended CTD[24], we hypothesize that despite significant sequence divergence these domains may have similar functions in TC-NER.

Our data indicate that the Elf1 interacts with the PH domain of the p62 subunit of TFIIH via a consensus TIR in its CTD. Mutation of a key aromatic residue in the TIR (*elf1*-Y99A), caused elevated UV sensitivity, a partial TC-NER defect in a *rad26Δ* mutant background, and a defect in PH domain binding in vitro, supporting this model. Our data also indicate that the UV sensitivity of the *elf1*-Y99A mutant is epistatic with a deletion in the p62 PH domain (*tfb1*-PHΔ), consistent with the model that Elf1 Y99 promotes UV resistance through its interaction with the PH domain. However, the *elf1*-Y99A phenotypes were not as severe as a complete CTD deletion, suggesting the hypothesis that other regions of the Elf1-CTD may also contribute to the TFIIH recruitment and TC-NER. This hypothesis may explain our preliminary CPD-seq data for the *elf1*-Y99A mutant in a *RAD26* WT background, which unlike the *elf1*-CTDΔ mutant did not have a significant TC-NER defect (Supplementary Fig. 13). This hypothesis is consistent with previous studies of the yeast GG-NER sensor Rad4, which revealed that disruption of the Rad4 TIR region alone caused only mild UV sensitivity unless the TIR mutant was combined with mutations in the Rad4 CTD, which was also important for NER and TFIIH binding[44]. It will be important in future studies to explore which residues in the Elf1-CTD, in addition to Y99, contribute to TFIIH binding and TC-NER.

The AlphaFold-multimer structure of the Elf1-CTD bound to the p62 PH domain (Fig. 6a) reveals a similar binding mode to that previously observed for other TIR regions[13,15,17,18]. The Elf1 Y99 residue inserts into the same pocket bound by UVSSA F408, Rad4 F95, and TFIIEα F387, aromatic residues that have also been shown to be important for PH binding. In the UVSSA, Rad4, and TFIIEα structures, a neighboring valine residue (i.e., UVSSA V411, Rad4 V98, or TFIIEα V390) inserts into a second pocket in the PH domain, but Elf1 lacks this

valine residue. This may explain why the binding affinity of the Elf1 CTD to the p62 PH domain is significantly weaker than the TIRs of UVSSA ($K_d = 71$ nM), Rad4 ($K_d = 50$ nM), and TFIIEα ($K_d = 45$ nM)[13,44,45]. However, the Elf1 CTD has a roughly similar binding affinity to p62 PH domain ($K_d = 3.82$ μM) as the unmodified TIRs in the transcription factors p53 ($K_d = 3.18$ μM to 24.21 μM)[15,46] and DP1 ($K_d = 35.3$ μM)[16], which are known to recruit TFIIH during transcription activation by binding the p62 PH domain. Like Elf1, the p53 and DP1 TIRs lack a neighboring valine residue to bind the second PH domain pocket, which can potentially explain the similarity in their binding affinities.

Our genetic data in yeast indicate that the PH domain mutant (*tfb1*-PHΔ) is more UV sensitive in a GG-NER deficient background than the *elf1*-CTDΔ or *elf1*-Y99A mutants (Supplementary Fig. 12). One interpretation of these findings is that other TIR-containing proteins may also function to bind the PH domain and recruit TFIIH during TC-NER. A likely candidate is Rad4, since previous studies have indicated that Rad4 is also required for TC-NER[47,48], and a recent study indicates that the Rad4 TIR regulates TC-NER in yeast through its interaction with the p62 PH domain[41]. This hypothesis may explain why the *elf1*-CTDΔ mutant does not completely abolish TC-NER in yeast, since Rad4 may serve as a functional replacement in binding and recruiting TFIIH. Consistent with this hypothesis, our preliminary data indicate that the Elf1-CTD shows enhanced UV sensitivity when mutated in combination with a *rad4*-TIRΔ mutant (deletion of Rad4 residues 85–102; see Supplementary Fig. 14), suggesting that they function in parallel and partially redundant pathways to recruit TFIIH during TC-NER. This mechanism may play a particularly important role in *rad16Δ* strains, since Rad4 presumably no longer functions to recruit TFIIH during GG-NER, and therefore can prioritize recruiting TFIIH during TC-NER. This model can explain previous observations indicating that TC-NER activity is elevated (and TC-NER defects partially mitigated) in *rad16Δ* cells[49,50], consistent with own *rad16Δ elf1*-CTDΔ CPD-seq data. Alternatively, the higher UV sensitivity of the PH domain mutant may reflect its role in subsequent steps of the NER reaction. For example, the Rad2 endonuclease, which makes the 3′ incision during NER[51], has been reported to interact with the PH domain of p62 and this interaction is important for NER in yeast[41,44,52]. The *elf1*-CTDΔ mutant is also less UV sensitive than a complete *ELF1* deletion. A recent report, which indicated that the core domain of Elf1 also facilitates TC-NER by facilitating the binding of Rad26 to lesion-stalled Pol II[53], can potentially explain this observation. These data, in conjunction with our findings reported herein, indicate that Elf1 promotes TC-NER by two distinct mechanisms, namely promoting initial recognition of stalled Pol II by Rad26 and facilitating subsequent TFIIH recruitment.

ELOF1, the human homolog of yeast Elf1, is also required to recruit TFIIH to stalled Pol II during TC-NER. However, unlike yeast Elf1, human ELOF1 lacks a CTD and is not thought to directly bind TFIIH (Fig. 6b). Instead, ELOF1 promotes recruitment of a second TC-NER factor known as UVSSA by facilitating ubiquitination of a single lysine residue (Rpb1 K1268) in the largest subunit of Pol II[19,20]. Once UVSSA is stably recruited to stalled Pol II by ELOF1, UVSSA subsequently binds the PH domain of p62 to recruit TFIIH[11,13]. This mechanism likely does not operate in budding yeast, since it (and many other species) lacks a UVSSA homolog and our preliminary data (Supplementary Fig. 15) indicate that mutations in the homologous lysine residue in yeast Pol II (i.e., *rpb1*-K1246R) do not impart UV sensitivity, consistent with a previous report[54]. Instead, we propose that the Elf1 CTD is the functional homolog of UVSSA since both contain a TIR that directly binds the same subunit of TFIIH.

Unlike UVSSA, the Elf1 CTD does not need to be recruited by Pol II ubiquitination, since Elf1 is a transcription elongation factor that is bound to Pol II[22–25,55]. Instead, we hypothesize that the activity of the Elf1 CTD must be regulated so that it only binds and recruits TFIIH when Pol II stalls at a DNA lesion. Notably, our in vitro binding data indicate that the Elf1 CTD is unable to bind TFIIH in the context of full-

length Elf1, suggesting that other domains in Elf1 may regulate its TFIIH binding activity, potentially as a mechanism to prevent spurious binding to TFIIH. It is also possible that other factors or post-translational modifications of Elf1 (e.g., phosphorylation[56]) serve as additional layers in regulating the selective binding of Elf1 CTD to TFIIH when a DNA lesion is present, and prevent its binding to TFIIH in the absence of DNA lesions during normal productive transcription elongation. Future studies will be required to understand the mechanism of this regulation.

In summary, our data reveal that a common structural mechanism, which operates to recruit TFIIH during transcription (Supplementary Fig. 16a) and GG-NER (Supplementary Fig. 16b), also mediates TFIIH recruitment by the Elf1-CTD during TC-NER in yeast (Fig. 6). That the human TC-NER factor UVSSA uses a similar mechanism to recruit TFIIH in mammalian cells[13] suggests that the Elf1 CTD and UVSSA likely have equivalent and conserved functions in TC-NER.

## Methods

### Strains and plasmids

All yeast strains were constructed in the BY4741 strain background. Complete gene deletions and 3xFLAG gene tagging were done using homologous recombination-based methods[57] and confirmed by PCR. Specific domain deletions (e.g., *elf1*-CTDΔ, *elf1*-NΔ, *tfb1*-PHΔ, *rad4*-TIRΔ) and/or point mutations in *ELF1* or *TFB1* were done using our published yeast CRISPR system[39,40] and confirmed by Sanger sequencing. For the plasmid-based expression of Spt4, we used a plasmid in which the promoter, coding sequence, and 3′ terminator sequences of *SPT4* were inserted into the multiple cloning sites of the *URA3* plasmid, pRS416[58] (gift from Shisheng Li, Louisiana State University, USA). Yeast strains and other materials are available upon request to the corresponding authors. Oligonucleotide sequences are given in Supplementary Data 1.

### Protein expression and purification

Expression and purification of yeast Elf1 and Elf1 truncations were performed essentially as previously described[23,55]. Briefly, GST-tagged Elf1 protein was expressed in *Escherichia coli* Rosetta 2(DE3) (Novagen, #71397) and purified by Glutathione Sepharose 4 Fast Flow resin (GE Healthcare, #17513202), and Superdex 200 10/300 GL column (GE Healthcare, #17517501). Expression and purification of the PH domain (1–115) of TFIIH p62 subunit was performed as previously described[59] with certain modifications. Briefly, GST-PH domain was expressed in host strain Rosetta 2(DE3), purified by Glutathione Sepharose 4 Fast Flow resin (GE Healthcare). After extensive wash, the PH domain was released from the resin by overnight PreScission protease cleavage.

### Yeast UV sensitivity test

The yeast cells were grown in Yeast extract Peptone Dextrose (YPD) medium at 30 °C overnight. For spotting assays, these cells were 10-fold serially diluted in fresh YPD medium with the first dilution containing ~1×10⁸ cells/ml and spotted on YPD plates. The plates were incubated at 30 °C in the dark after exposing to different doses of UVC light (254 nm), based on our previous calibration, and images were taken after 3–5 days of incubation. For quantitative UV survival assay, diluted yeast cells were plated on YPD plates and exposed to the indicated UV dose, based on our previous calibration. The number of colonies was counted after incubating the plates for 3 days at 30 °C in the dark. The survival graph depicts the mean and SEM of three independent experiments.

### CPD-seq library preparation and sequencing

The WT or the mutant yeast cultures were grown to an OD$_{600}$ of ~0.8–1.0, pelleted, and re-suspended in dH$_2$O. Cells were collected for a "No UV" sample, and the remaining cells were irradiated with 125 J/m² UVC light (254 nm), based on previous calibration. Cells were collected immediately after UV exposure (0 h) and the remaining cells after UV treatment were incubated in the dark in pre-warmed fresh YPD medium for repair. The cells were collected after 2 h (2 h) repair incubation at 30 °C. The cells were pelleted and stored in the −80 °C freezer until genomic DNA isolation. Genomic DNA extraction was done using phenol:chloroform:isoamylalcohol (PCI) extraction and ethanol precipitation. The CPD-seq library preparation and quality control, sequencing with an Ion Proton sequencer, and data processing were performed essentially as previously described[27,28]. Briefly, genomic DNA was sonicated (30 s ON/OFF, 25 cycles; Diagenode Bioruptor 300) to an average size of ~400 bp, end-repaired (NEB, E6050L), and dA-tailed (NEB, E6053L) using the indicated NEB kits. DNA was then ligated to the double-stranded adapter DNA trP1 using Quick ligase (NEB, E6056L), and treated with terminal transferase (NEB, M0315L). DNA was then digested with T4 endonuclease V (T4 PDG, NEB, M0308S) and AP endonuclease (NEB, M0282S) to create 3′-OH groups immediately upstream of the CPD lesions, followed by treatment with Shrimp alkaline phosphatase (Thermo Fisher Scientific, 78390500UN). The resulting DNA fragments were ligated to a biotin-labeled second adapter (adapter A). Up to 6 different barcoded A adapters (barcode sequences are A1: AAGAGGAT; A2: TTCGTGAT; A3: CCTGAGAT A4: ATCGCGAT; A5: TACTGGAT; and A6: GAACTGAT) were used to generate multiplexed libraries for different experiments. Fragments ligated to the A adapter were selected for with streptavidin beads (Thermo Fisher Scientific, 11205D) with a high affinity for biotin on the second adapter. These fragments were PCR amplified and sent out for Ion Torrent sequencing. Ampure XP beads (Beckman Coulter, A63881) were used for size selection and cleanup between enzymatic steps.

The resulting sequencing reads were trimmed of the barcode (and typically one nucleotide at the 3′ end of the read), aligned to the yeast genome (SacCer3) using bowtie2[60], and processed using SAMtools[61] and BEDTools[62] to generate a BED file. CPD-seq reads associated with lesions at dipyrimidine sequences (i.e., TT, TC, CT, CC) were retained for further analysis. BED files were split into individual DNA strands and converted to wig files for subsequent analysis. Dipyrimidine sequences associated with no CPD-seq reads were assigned a read count of 0. CPD-seq data for WT, *rad26*Δ, *rad16*Δ controls are from[19,30,32,33]. Data processing for these data sets was generally performed as described previously[29]. For transcriptional asymmetry analysis, the fraction of CPDs remaining in each bin for each DNA strand was averaged for the two *rad16*Δ[33] or *rad26*Δ[30] replicates prior to calculating log$_2$ TS/NTS ratio. The other *rad16*Δ replicate (i.e., Supplementary Fig. 5b) is derived from a control anchor-away strain in a W303 background treated with rapamycin[32].

### CPD-seq data analysis

Analysis of CPD repair around the transcription start site (TSS) of ~5200 yeast genes was performed as previously described[19,29,30,32,33]. Briefly, custom Perl scripts analyzed the number of CPD lesions at each position on the TS and NTS adjacent to the TSS positions for ~5200 yeast genes obtained from ref. 63, both for the 2 h repair and 0 h control. The fraction of CPDs remaining in the 2 h CPD-seq data relative to the 0 h control at each position relative to the TSS was plotted. For the *elf1*-CTDΔ and WT control data sets, the fraction of CPDs remaining in all analyses was normalized using the fraction of unrepaired CPDs after 2 h repair determined by alkaline gel electrophoresis. Positioning of nucleosome dyad positions relative to the TSS was analyzed using published MNase-seq data[64], as previously described[29,30,32,33]. Bin plot analysis of repair between the TSS and TES was performed similarly using custom Perl scripts, as previously described[19,29,30,32,33], except that a modified Perl script that fixed a small error related to bin boundaries was used to perform bin plots of individual yeast genes. Briefly, each of ~5000 yeast genes that have coordinates for the TSS and TES (derived from position of poly-adenylation site (PAS) coordinates[63]) were divided into six equally

sized bins, dependent upon gene length, and the number of CPDs in each bin was counted. Three additional bins uniformly 167 bp in length were also analyzed upstream of the TSS and downstream of the TES of each yeast gene. The fraction of CPDs remaining after 2 h repair relative to the 0 h control was plotted for all yeast genes, and for individual yeast genes, which were visualized using the Java TreeView software[65]. The $\log_2$ ratio of the fraction of CPDs remaining in the TS and NTS was also plotted. TES coordinates were derived from the published location of the PAS for each yeast gene[63]. Finally, the average $\log_2$ ratio of CPDs remaining in the TS and NTS was calculated using the average of $\log_2$ TS/NTS ratio of the six transcribed bins between the TSS and TES of yeast genes.

## Global analysis of CPD repair in yeast by alkaline gel electrophoresis

The global repair of CPD lesions was analyzed by T4 endonuclease V digestion and alkaline gel electrophoresis, as described previously[19,29,66]. In brief, WT or *elf1-CTDΔ* mutant strains were grown in YPD to mid-log phase ($OD_{600}$ ~ 0.65), pelleted, and re-suspended in $dH_2O$. Cells were collected for a "No UV" control and the rest of the cells were exposed to 125 J/m$^2$ UVC (254 nm). The UV-exposed cells were pelleted, re-suspended in YPD medium, and incubated at 30 °C with aliquots taken at different timepoints as indicated. Genomic DNA was isolated from these samples using PCI extraction and ethanol precipitation. Equal amounts of DNA (50 μg) were treated with T4 endonuclease V/PDG (NEB M0308S) and the digested samples were resolved using alkaline gel electrophoresis. The gel was subsequently stained with SYBR Gold (Invitrogen), imaged with Typhoon FLA 7000 (GE Healthcare), and quantified with ImageQuant 5.2. Graphs represent the mean and SEM of three independent experiments.

## Western blot

Cells were cultured in YPD medium at 30 °C to O.D.600 ~ 1 and whole cell protein extracts were prepared from the cells using the procedure described previously[67]. The protein extracts were analyzed by western blot using antibodies against total histone H3 (anti-H3, Abcam, ab46765) and FLAG (anti-FLAG, Sigma-Aldrich, F1804) epitope. Both antibodies were used at a 1:10,000 dilution. Blots were scanned using Typhoon FLA 7000 (GE Healthcare).

## In vitro pull-down assay

The quantities of the purified protein were assessed by Bradford assay. GST Beads containing estimated equivalent quantities of the tagged Elf1 protein truncations were mixed with the purified PH domain of TFIIH p62 subunit. The mixtures were incubated at 23 °C for 1 h with rotation. After centrifugation, the supernatants were removed, and the beads were washed twice in a buffer composed of 20 mM HEPES, pH 7.4, 5% glycerol. Interacting proteins were analyzed by SDS-PAGE.

## Quantitative binding assays using Biolayer Interferometry

To elucidate the impact of the Elf1-Y99 residue on the binding affinity within the PH domain of p62, we quantified the dissociation constant ($K_d$) for both wild-type and Y99A mutant forms of Elf1-CTD (residues 85–145) using Biolayer Interferometry (BLI). The Octet® K2 system was utilized for these measurements, adhering strictly to the manufacturer's guidelines. In brief, recombinant Elf1-CTD proteins (Supplementary Fig. 17; uncropped gel and blot images are in Source Data), both wild-type and Y99A mutant, tagged with glutathione S-transferase (GST), were immobilized onto GST-specific biosensors. Subsequently, the PH domain of p62 was serially diluted in BLI assay buffer (20 mM Tris-HCl, pH 7.5, 5% glycerol, and 1 mM DTT) to the following concentrations: 30 μM, 15 μM, 7.5 μM, 3.75 μM, and 0 μM. Each biosensor with bound Elf1-CTD was exposed to a different concentration of the PH-p62 protein for an association phase of 2 min, followed by a dissociation phase in BLI buffer for 3 min. This procedure was replicated in three independent experiments to ensure the reliability of the $K_d$ values obtained. The response signals generated during the assay were automatically captured and processed by the Octet system's software (Data Acquisition v12.0.2.11 and Data Analysis HT v12.0.2.59 from FORTEBIO), which was also used for the subsequent analysis and calculation of the $K_d$ values.

## Reporting summary

Further information on research design is available in the Nature Portfolio Reporting Summary linked to this article.

## Data availability

The CPD-seq data generated in this manuscript have been deposited in NCBI's Gene Expression Omnibus (GEO) database and are available under GEO series accession code GSE243603. The published CPD-seq data for WT, *rad26Δ*, and *rad16Δ* is available from GEO with the accession numbers: GSE161930 GSE145911 GSE149082 GSE131101. Source data are provided with this paper.

## Code availability

The custom Perl scripts used in the study have been deposited at GitHub: https://github.com/bmorledge-hampton19/ELF1_CTD[68].

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

## Acknowledgements

We are grateful to Dr. Shisheng Li for providing the *pRS416-SPT4* plasmid. We thank Benjamin Morledge-Hampton for assistance in depositing the Perl scripts. This work was supported by grants from the National Institute of General Medical Sciences (R01 GM102362 to D.W.) and the National Institute of Environmental Health Sciences (R01ES028698 (J.J.W.), R01ES032814 (J.J.W.), and R21ES035139 (J.J.W.)).

## Author contributions

K.S., J.X., H.E.W., D.W., and J.J.W. designed the research and interpret results; K.S. performed the yeast UV sensitivity and CPD-seq experiments; J.X., J.O., and Q.L. purified proteins and perform in vitro pull-down and binding assays. K.S. and J.J.W. analyzed the CPD-seq data; H.E.W. performed the alkaline gel repair studies; K.S., J.X., H.E.W., J.O., Q.L., D.W., and J.J.W. wrote and edited the paper.

## Competing interests

The authors declare no competing interests.
