## [Peer Review File · Nature Communications]

Elf1 promotes transcription-coupled repair in yeast by using its C-terminal domain to bind TFIIDREVIEWER COMMENTS

Reviewer #1 (Remarks to the Author):

Elf1/Elof1 is a newly identified TCR factor in both yeast and mammalian cells. In the latter case, Elof1 promotes TCR by stimulating UVSSA mono-ubiquitination. However, yeast lacks UVSSA homolog, thus the role of Elf1 in yeast is unknown. In this manuscript, Selvam et al. show that the CTD of Elf1 contains a TFIIH-interaction region (TIR) similar to TIR of mammalian UVSSA and XPC and yeast Rad4 (the yeast homolog of XPC). They demonstrate that Elf1 CTD can bind to the PH domain of TFIIH p62 subunit, and this domain as well as its interaction with TFIIH is important for both Rad26-dependent and Rad26-independent TCR in yeast. This study suggests that yeast Elf1 is involved in TCR by recruiting TFIIH. Overall, the viewpoint of this paper is novel and logical, and the data is well presented and can support the point.

My points of concern:

1. TFIIH recruitment is an indispensable step of NER. I note that there is a significant level of Rad26-dependent TCR left in either *elf1Δ* (Geijer et al. Nature Cell Biology 2021) or *elf1-CTDΔ* cells, while loss of UVSSA can completely abolish mammalian TCR. How are TFIIH recruited in the absence of Elf1 or its CTD? Since Rad4 is also required for TCR in yeast and contains a TIR motif, does it play a role in TFIIH loading in TCR? If it does, how are Elf1 and Rad4 coordinated during TCR?
2. It seems that there is no Rad26-independent TCR in the absence of Elf1-CTD. Is it true or just due to the sensitivity of CPD-seq that cannot detect very weak TCR? The *elf1-Y99* also plays a more important role in Rad26-independent TCR than in Rad26-dependent TCR. If Elf1-CTD is indispensable or more important in Rad26-independent TCR, what is the possible explanation? If the Rad26-independent TCR in the absence of Elf1-CTD is just too weak, please try to detect it. Deleting *rad16* may enhance the signal of TCR, although the *rad16Δrad26Δelf1Δ* or *rad16Δrad26Δelf1-CTDΔ* strain is very sensitive to UV. Optimization of UV dose/repair time as well as alternative methods are required to detect the possible weak repair in the triple-mutant cells. Indeed, the *rad16Δrad26Δelf1Δ* strain showed weak repair in the template strand of the RPB2 locus, although the repair of non-template strand was not presented (Geijer et al. Nature Cell Biology 2021, Extended Data Fig. 5 g and h).

Reviewer #2 (Remarks to the Author):

Transcription-coupled repair is not well understood and may proceed via multiple pathways. A lack of perfect correspondence between TCR factors in different species makes it more difficult. Here the authors present results suggesting that the C-terminal domain of yeast Elf1 acts as a functional homologue of the human UVSSA protein. The authors reveal a key role of the Elf1 CTD in Rad26-independent transcription-coupled repair using genetic and genome-wide studies that complement one-another well. They show that strains lacking the Elf1-CTD have a strand-asymmetric repair pattern – a hallmark of transcription-coupled repair. In addition, they identify a key residue and region in the Elf1 CTD that is important for TFIIH binding. These data serve as a foundation for more in-depth mechanistic studies. Mostly the conclusions drawn by the authors are supported well by their data and I recommend this paper for publication upon minor revisions. My criticisms and questions are laid out below:

1. Introduction I – it would be appropriate to mention the role of Elf1 in transcription elongation (independent of TCR) here. This important fact doesn't show up until later in the paper.
2. Introduction II and Results p10 – the authors state that Elf1 plays a role in both Rad26-dependent and Rad26-independent TCR. The use of "Rad26-independent TCR" throughout the paper is justified since you often probe the role of Elf1 in cells lacking Rad26, but I don't see the evidence for Elf1 playing a specific-role connected to Rad26 function.
3. Results section I – it would be helpful to specify that "Rad26-independent repair" is transcription-coupled, for readers that may be unsure whether Rad26-independent repair means a version of TCR or simply just GGR.

4. Results section II – could the authors explain why 5,200 genes are studied sometimes and 5,000 genes are studied other times? What was the criteria for excluding genes in some cases?
5. Results section III – you nicely show that the Elf1 CTD is required for efficient TS repair for most genes, yet in the bulk assay mentioned at the start of this section you don't see a significant effect of the Elf1-CTDΔ mutant – is this because GGR is the predominant pathway in cells? (The Rad16 deletion seems pretty resistant to uv.) It would be helpful to comment on this interesting discrepancy, as recent results from bacteria show that GGR is a very minor contributor to genome stability, whilst TCR is carrying out most of the repair of such lesions (Martinez et al. Nat Comm 2022).
6. Results section III – "CPDS" should be CPDs
7. Results section III – what's the rationale for using a log2 ratio in these analyses?
8. Results section III – terms like "closer to zero" are vague. Better if they were replaced by numbers with associated errors.
9. Results section V – it may be helpful to have a summary for all of the CPD-seq data, either here or in the discussion, it is all very interesting, but a brief summary of the key findings would orient the reader well for the discussion.
10. Discussion – the authors show earlier that full-length Elf1 is unable to bind the PH domain of TFIIH, indicating potential auto-inhibition – could the authors speculate as to how this might be helpful during transcription in the absence of damage vs the presence of damage?
11. Discussion – the authors state that "Elf1 is a transcription elongation factor that is constitutively bound" yet the references are to mostly structural studies that don't provide information on Elf1 lifetime nor dynamics. We simply don't know if Elf1 is always bound to RNA pol II so the text should be altered to reflect this.
12. Discussion – "binding of Elf1 CTD to TFIIH in the when a DNA lesion is present" is a type.

Reviewer #3 (Remarks to the Author):

The yeast Elf1 and its human ortholog ELOF1 have previously been found to play an important role in transcription coupled nucleotide excision repair (TC-NER). This manuscript by Selvam et al presented convincing data showing that the C-terminal domain (CTD) of Elf1, which is absent from the human ELOF1 sequence, plays a significant role in TC-NER. The authors also show that the Elf1 CTD co-pulldown with the pleckstrin homology (PH) domain of the p62 subunit of TFIIH in vitro. The findings are potentially significant, as how TFIIH is recruited during TC-NER in yeast remains unknown. However, the experimental data presented in the manuscript does not appear to be sufficient to conclusively support the overarching assertion that TFIIH is recruited through interaction with Elf1 CTD during TC-NER. Additional experiments are necessary to fortify this central claim.

Major concerns:

1. The absence of structural data illustrating the in vitro interaction between Elf1 CTD and p62 PH is a notable limitation. Co-structures involving p62 PH with various proteins, such as UVSSA and XPC, have been documented. The lack of this structural information raises questions about the validity of the interaction and the assertion of the similarity between yeast Elf1 and human UVSSA in recruiting TFIIH.
2. While the manuscript indicates a p62 PH co-pulldown with GST-tagged Elf1 fragments in vitro, essential details are absent. Notably, there is no information on the binding affinity of Elf1 CTD fragments with p62 PH, and the impact of the Elf1 Y99A mutation on binding affinity remains undisclosed.
3. The manuscript falls short in providing evidence that Elf1 CTD promotes TC-NER through interaction with p62 PH in cellular contexts. Data supporting the functional relevance of this interaction within the cell are notably absent, limiting the substantiation of the proposed mechanism for Elf1 CTD in TC-NER.

Minor points:

1. All the data and schematic in Figure 1 have been published in previous publications. It is better not

to show them, or at most show them in the Supplementary Materials.

2. Figure 2 C-E. The major finding in the manuscript is the role of Elf1 CTD in TC-NER. What is the reason for the lack of impact of *elf1*-CTD Δ on overall NER, encompassing both TC- and GG-NER? The assertion that the *elf1*-CTD Δ does not affect GG-NER solely based on the overall NER gel is insufficient. What's the rationale underlying this assertion? An asterisk * is missing in the plot (E).

3. Figure 2F-G. From the plots, it appears that GG-NER is faster in *elf1*-CTD Δ than in WT. The manuscript states that "The fraction of CPDs remaining was normalized to the bulk fraction of CPDs remaining determined by alkaline gel analysis". This normalization appears to have an intrinsic problem, as it cannot independently normalize the lesions repaired by GG-NER and TC-NER.

4. Figure 3E is identical to Figure 1C. While recognizing the intention to compare the data of *elf1*-CTD Δ with *rad26* Δ here. However, presenting the same data in different figures does not appear to be necessary.

5. UV sensitivity and TC-NER data of *elf1* Δ , *elf1*-CTD Δ and *elf1*-Y99A are presented in different genetic backgrounds (WT, *rad16* Δ , *rad26* Δ and *rad16* Δ *rad26* Δ) across figures. This way of presentation makes it hard for readers to assess the degrees of TC-NER defects caused by different *elf1* mutants in different genetic backgrounds. The authors may consider presenting UV-sensitivity data in one multi-panel figure and repair data in another to enhance clarity for readers.

RESPONSE TO REVIEWERS

Reviewer #1:

1. “TFIIH recruitment is an indispensable step of NER. I note that there is a significant level of Rad26-dependent TCR left in either *elf1* Δ (Geijer et al. Nature Cell Biology 2021) or *elf1*-CTD Δ cells, while loss of UVSSA can completely abolish mammalian TCR. How are TFIIH recruited in the absence of Elf1 or its CTD? Since Rad4 is also required for TCR in yeast and contains a TIR motif, does it play a role in TFIIH loading in TCR? If it does, how are Elf1 and Rad4 coordinated during TCR?”

RESPONSE: We thank the Reviewer for this interesting question. While our data indicate that the Elf1 CTD contributes to TFIIH recruitment, deletion of Elf1 or its CTD does not completely eliminate TC-NER. These data indicate that other factors contribute to TFIIH recruitment, particularly in the absence of Elf1. One possible candidate is Rad4, as suggested by the Reviewer, since Rad4 in yeast is required not only for GG-NER, but also TC-NER. We previously tried to analyze CPD repair genome-wide in a *rad4* Δ mutant, but were unable to generate CPD-seq libraries due to apparent technical issues. A second possibility is Rpb9, since it is required for Rad26-independent TC-NER. However, there is no evidence linking Rpb9 to TFIIH recruitment. A third possibility is Def1, which binds to Rad26¹ and TFIIH², and plays a role in RNA polymerase II ubiquitination and degradation¹. We have preliminary data that Def1 plays genome-wide role in TC-NER, potentially by binding the p62 pleckstrin homology (PH) domain in TFIIH, which we plan to publish in a future study.

Hence, we hypothesize that Def1 (and potentially Rad4) may function to recruit the p62 PH domain in TFIIH in the absence of Elf1. Consistent with this hypothesis, we now include new data (see Supplementary Fig. 12) showing that p62 PH domain mutant (*tfb1*-PH Δ) shows greater UV sensitivity in a *rad16* Δ mutant background than an *elf1*-CTD Δ mutant, indicating that other factors (e.g., Def1, Rad4) may also interact with the PH domain during TC-NER. We now discuss these important questions in the discussion section of the revised manuscript (see pages 18-19).

2. “It seems that there is no Rad26-independent TCR in the absence of Elf1-CTD. Is it true or just due to the sensitivity of CPD-seq that cannot detect very weak TCR? The *elf1*-Y99 also plays a more important role in Rad26-independent TCR than in Rad26-dependent TCR. If Elf1-CTD is indispensable or more important in Rad26-independent TCR, what is the possible explanation? If the Rad26-independent TCR in the absence of Elf1-CTD is just too weak, please try to detect it. Deleting *rad16* may enhance the signal of TCR, although the *rad16* Δ *rad26* Δ *elf1* Δ or *rad16* Δ *rad26* Δ *elf1*-CTD Δ strain is very sensitive to UV. Optimization of UV dose/repair time as well as alternative methods are required to detect the possible weak repair in the triple-mutant cells. Indeed, the *rad16* Δ *rad26* Δ *elf1* Δ strain showed weak repair in the template strand of the RPB2 locus, although the repair of non-template strand was not presented (Geijer et al. Nature Cell Biology 2021, Extended Data Fig. 5 g and h).”

RESPONSE: Our CPD-seq data (Fig. 3) suggest that the deletion of the Elf1-CTD eliminates most of the apparent Rad26-independent repair in GG-NER proficient cells. It is certainly possible that the CPD-seq method is not sensitive enough to detect low levels of ongoing Rad26-independent TC-NER. As an alternative approach to address this question, we compared the UV sensitivity of *rad26Δ elf1-CTDΔ* mutant cells in a *rad16Δ* background relative to *rad16Δ rad26Δ rpb9Δ*, as previous studies indicate that this mutant is completely defective in Rad26-independent TC-NER³. These new data indicate that while *elf1-CTDΔ* enhances the sensitivity of a *rad16 rad26* double mutant to relatively higher UV doses (e.g., 7.5 J/m², see Supplementary Fig. 6), consistent with our previous results, it causes significantly less UV sensitivity than the *rad16Δ rad26Δ rpb9Δ* mutant (Supplementary Fig. 6). Taken together, these data suggest that the *elf1-CTDΔ* does not completely eliminate Rad26-independent repair, at least in a *rad16Δ* mutant background, but does significantly diminish Rad26-independent repair. These new findings are discussed on pages 10-11 of the revised manuscript.

Reviewer #2:

1. "Introduction I – it would be appropriate to mention the role of Elf1 in transcription elongation (independent of TCR) here. This important fact doesn't show up until later in the paper."

RESPONSE: We now mention the role of ELOF1/Elf1 in transcription elongation in the introduction section on page 4, paragraph 2 of the revised manuscript.

2. "Introduction II and Results p10 – the authors state that Elf1 plays a role in both Rad26-dependent and Rad26-independent TCR. The use of "Rad26-independent TCR" throughout the paper is justified since you often probe the role of Elf1 in cells lacking Rad26, but I don't see the evidence for Elf1 playing a specific-role connected to Rad26 function."

RESPONSE: We agree and have modified the text here and elsewhere (e.g., see pages 5, 7, etc. of the revised manuscript) to indicate instead that Elf1 and the Elf1 CTD function in TC-NER in both wild type and *rad26Δ* mutant cells.

3. "Results section I – it would be helpful to specify that "Rad26-independent repair" is transcription-coupled, for readers that may be unsure whether Rad26-independent repair means a version of TCR or simply just GGR."

RESPONSE: We have modified the text here and elsewhere to read instead "Rad26-independent TC-NER."

4. “Results section II – could the authors explain why 5,200 genes are studied sometimes and 5,000 genes are studied other times? What was the criteria for excluding genes in some cases?”

RESPONSE: Coordinates describing the position of the transcription start site (TSS) is available for ~5200 yeast genes. That is why repair of ~5200 genes is analyzed around the TSS. However, only 5000 genes also have corresponding coordinates for the transcription end site (TES – corresponding to the polyadenylation site). That is why analysis of repair from the TSS to TES (i.e., Supplemental figures 4 and 5) only analyzed 5000 genes. More details are provided in the revised method section (pages 23-24).

5. “Results section III – you nicely show that the Elf1 CTD is required for efficient TS repair for most genes, yet in the bulk assay mentioned at the start of this section you don’t see a significant effect of the Elf1-CTD Δ mutant – is this because GGR is the predominant pathway in cells? (The Rad16 deletion seems pretty resistant to uv.) It would be helpful to comment on this interesting discrepancy, as recent results from bacteria show that GGR is a very minor contributor to genome stability, whilst TCR is carrying out most of the repair of such lesions (Martinez et al. Nat Comm 2022).”

RESPONSE: TC-NER operates only on transcribed DNA, which comprises only a subset of the yeast genome. Moreover, GG-NER in yeast occurs very efficiently, such that nearly all CPD lesions are repaired within 3 to 4 hours after UV irradiation, and therefore GG-NER is the predominant repair pathway in these cells, as suggested by the Reviewer. Because defects in TC-NER affect only a subset of the yeast genome and can be readily compensated for by efficient GG-NER, loss of TC-NER in yeast is only expected to have a minor effect on overall CPD repair in bulk DNA. Consistent with this hypothesis, mutants in TC-NER genes (e.g., *rad26* Δ and *elf1* Δ) show very mild UV sensitivity phenotypes on their own, which is why UV sensitivity assays in yeast often are performed in a GG-NER deficient mutant background (i.e., *rad16* Δ). This point is mentioned on page 6, paragraph 2 and page 8, paragraph 1 of the revised manuscript.

6. “Results section III – “CPDS” should be CPDs”

RESPONSE: This typo has been corrected.

7. “Results section III – what’s the rationale for using a log₂ ratio in these analyses?”

RESPONSE: By calculating the log₂ ratio of the fraction of CPDs remaining on the TS relative to the NTS, we can directly quantify the degree of repair asymmetry between the two DNA strands, independent of other variables, such as overall repair rate, etc., since these variables should contribute equally to both the numerator and denominator of the ratio. Taking the log₂ of this ratio makes the resulting statistic symmetric, in the

sense that a 2-fold decrease has the same magnitude of log₂ ratio (i.e., -1) as a two-fold increase (log₂ ratio = +1). The log₂ ratio statistic was used in the first paper analyzing transcriptional asymmetry (in this case of mutations)⁴, and we have used it in our previous studies analyzing TC-NER in yeast^{5, 6}.

8. “Results section III – terms like “closer to zero” are vague. Better if they were replaced by numbers with associated errors.”

RESPONSE: We agree, and have now included numbers quantifying the average transcriptional asymmetry (i.e., log₂ ratio of unrepaired CPDs on TS/NTS) for all six transcribed bins in yeast genes (see pages 9,10, and 14 of the revised manuscript).

9. “Results section V – it may be helpful to have a summary for all of the CPD-seq data, either here or in the discussion, it is all very interesting, but a brief summary of the key findings would orient the reader well for the discussion.”

RESPONSE: In the revised manuscript, we now include a brief summary of the CPD-seq and other data on page 16, as suggested by the Reviewer.

10. “Discussion – the authors show earlier that full-length Elf1 is unable to bind the PH domain of TFIIH, indicating potential auto-inhibition – could the authors speculate as to how this might be helpful during transcription in the absence of damage vs the presence of damage?”

RESPONSE: This is an interesting point. If full-length Elf1 is unable to bind the PH domain of TFIIH due to auto-inhibition, this might serve potentially important functions in the cell. First, it might prevent free Elf1 (i.e., not associated with RNA polymerase II) from binding to and potentially sequestering TFIIH. In this model, incorporation of Elf1 into the RNA polymerase II elongation complex might alleviate auto-inhibition. A second model is that Elf1 auto-inhibition may prevent the Elf1 CTD from prematurely recruiting TFIIH in the absence of damage. In this model, RNA polymerase II stalling at a lesion may alleviate auto-inhibition, thereby releasing the Elf1 CTD to recruit TFIIH. We now discuss these ideas on page 20, paragraph 2 of the revised manuscript.

11. “Discussion – the authors state that “Elf1 is a transcription elongation factor that is constitutively bound” yet the references are to mostly structural studies that don’t provide information on Elf1 lifetime nor dynamics. We simply don’t know if Elf1 is always bound to RNA pol II so the text should be altered to reflect this.”

RESPONSE: We now cite and discuss studies that utilized ChIP or ChIP-chip data in yeast to show that Elf1 is localized to the transcribed regions of yeast genes, and that its binding profile shows significant similarity to other elongation factors and RNA Pol II itself^{7, 8}. Importantly, Elf1 binding to the coding region of a model yeast gene is

transcription dependent⁷, consistent with its hypothesized role as an elongation factor. We have modified the text accordingly on page 4, paragraph 2 and page 20 of the revised manuscript.

12. “Discussion – “binding of Elf1 CTD to TFIIH in the when a DNA lesion is present” is a typo.”

RESPONSE: This typo has been corrected in the revised manuscript.

Reviewer #3:

0. “The findings are potentially significant, as how TFIIH is recruited during TC-NER in yeast remains unknown. However, the experimental data presented in the manuscript does not appear to be sufficient to conclusively support the overarching assertion that TFIIH is recruited through interaction with Elf1 CTD during TC-NER. Additional experiments are necessary to fortify this central claim.”

RESPONSE: We now include new data describing the binding affinity between the GST-Elf1 CTD and the purified p62 PH domain, and genetic studies indicating that the Elf1 CTD and Elf1 Y99 show epistatic interactions with the p62 PH domain in yeast, highlighting the functional relevance of this interaction and indicating that they function in the same cellular pathway of UV resistance. These new data are detailed below.

Major concerns:

1. “The absence of structural data illustrating the in vitro interaction between Elf1 CTD and p62 PH is a notable limitation. Co-structures involving p62 PH with various proteins, such as UVSSA and XPC, have been documented. The lack of this structural information raises questions about the validity of the interaction and the assertion of the similarity between yeast Elf1 and human UVSSA in recruiting TFIIH.”

RESPONSE: We now include an AlphaFold-multimer model (see Fig. 6a) of the binding of the putative TFIIH-interacting region of the Elf1 CTD with the p62 PH domain. In this predicted structure, Elf1 Y99 binds to the same pocket of the pH domain as TFIIH-interacting residues F408 (UVSSA), F95 (Rad4) and F387 (TFIIE) in previously solved structures. Although this is a predicted structure, the high accuracy of AlphaFold in predicting structures, in combination with our biochemical and genetic data, support the proposed model. We now discuss this new structural analysis on page 15, paragraph 2 of the revised manuscript.

2. “While the manuscript indicates a p62 PH co-pulldown with GST-tagged Elf1 fragments in vitro, essential details are absent. Notably, there is no information on the binding affinity of Elf1 CTD fragments with p62 PH, and the impact of the Elf1 Y99A mutation on binding affinity remains undisclosed.”

RESPONSE: We used Biolayer interferometry (BLI) to measure the dissociation constant (K_d) values for the WT Elf1-CTD or Y99A mutant in complex with the TFIIH p62 PH domain. These experiments yielded a K_d value of 3.82 μ M for WT Elf1-CTD, which is roughly similar to K_d values previously determined for p62 PH domain-binding transcription factors p53 ($K_d = 3.18 \mu$ M to 24.21 μ M)^{9, 10} and DP1 ($K_d = 35.3 \mu$ M)¹¹. Notably, these experiments also indicated that the K_d value of the Elf1 Y99A CTD mutant ($K_d = 7.32 \mu$ M) was significantly weaker than that of wild type ($P < 0.0001$). These new data are included in Fig. 5F and in Supplemental Tables 1 & 2 and Supplementary Figure 7, and are discussed on page 11,13 of the revised manuscript.

3. The manuscript falls short in providing evidence that Elf1 CTD promotes TC-NER through interaction with p62 PH in cellular contexts. Data supporting the functional relevance of this interaction within the cell are notably absent, limiting the substantiation of the proposed mechanism for Elf1 CTD in TC-NER.

RESPONSE: We now include new genetic data indicating that the Elf1 CTD, and Elf1 Y99 in particular, function in the same pathway as the PH domain of the p62 subunit of TFIIH. Deletion of the PH domain of p62 in yeast (i.e., *tfb1-PH* Δ) results in elevated UV sensitivity in both *rad16* Δ and *rad16* Δ *rad26* Δ mutant backgrounds, consistent with the PH domain playing an important role in TC-NER. Importantly, the *elf1*-Y99A mutant does not increase the UV sensitivity of the *tfb1-PH* Δ mutant in either the *rad16* Δ or *rad16* Δ *rad26* Δ mutant backgrounds (Supplementary Fig. 12). These data indicate that Elf1-Y99 is epistatic with the PH domain, and therefore likely functions in the same pathway during TC-NER. Similar analysis indicates that the Elf1-CTD is also largely epistatic to the p62 PH domain (Supplementary Fig. 12). Taken together, these findings functionally link the Elf1 CTD, and the Y99 residue in particular, to the p62/TFB1 PH domain in TC-NER. These new findings are discussed on pages 14-15 of the revised manuscript.

Minor points:

1. "All the data and schematic in Figure 1 have been published in previous publications. It is better not to show them, or at most show them in the Supplementary Materials."

RESPONSE: Only the data depicted in panels c,d in figure 1 (i.e., *rad26* Δ data) have been previously published, and are included for comparison purposes. The CPD-seq data for the *rad26* Δ *elf1* Δ mutant (see Fig. 1b,e,f) have not been previously published and these data show for the first time that Elf1 plays a role in Rad26-independent TC-NER throughout the yeast genome. The CPD-seq schematic shown in Figure 1a, while related to previously published schematic figures, is modified and distinct for this manuscript. We feel it is important to include the schematic to provide readers unfamiliar with the CPD-seq method a graphical description of how the method works. For these reasons, we have retained Figure 1 in the revised manuscript.

2. “Figure 2 C-E. The major finding in the manuscript is the role of Elf1 CTD in TC-NER. What is the reason for the lack of impact of *elf1*-CTD Δ on overall NER, encompassing both TC- and GG-NER? The assertion that the *elf1*-CTD Δ does not affect GG-NER solely based on the overall NER gel is insufficient. What’s the rationale underlying this assertion? An asterisk * is missing in the plot (E).”

RESPONSE: Previous studies have revealed that GG-NER occurs very rapidly in yeast (much more rapidly than in human cells), such that nearly all CPD lesions are repaired within 3 to 4 hours after UV irradiation. This efficient GG-NER therefore can readily compensate for defects in TC-NER. Consistently, mutations in genes involved in TC-NER typically have only mild sensitivity to UV light, unless tested in a GG-NER deficient (i.e., *rad16* Δ) mutant background. Moreover, TC-NER occurs on only one of the DNA strands (i.e., the transcribed strand) of yeast genes, which further comprise only a subset of the genome. For these reasons, the partial TC-NER defect observed in *elf1*-CTD mutants should have only a minimal effect on bulk repair of CPD lesions across the genome, consistent with our alkaline gel data. In contrast, our previous studies indicate that mutants that affect GG-NER typically cause significant differences in bulk repair of CPD lesions in yeast, as detected by T4 endo V digestion and alkaline gel analysis (e.g., ^{5, 12}). That we do not observe such a difference in bulk repair of CPDs by alkaline gel analysis in the *elf1*-CTD Δ mutant is consistent with the hypothesis that the Elf1-CTD does not affect GG-NER. This finding is also consistent with previously published results analyzing the role of Elf1 in repair in yeast¹³. Note, in Fig. 2e, none of the time points showed a significant difference in repair, which is why no asterisk (*) is depicted in the plot.

3. “Figure 2F-G. From the plots, it appears that GG-NER is faster in *elf1*-CTD Δ than in WT. The manuscript states that “The fraction of CPDs remaining was normalized to the bulk fraction of CPDs remaining determined by alkaline gel analysis”. This normalization appears to have an intrinsic problem, as it cannot independently normalize the lesions repaired by GG-NER and TC-NER.”

RESPONSE: The difference in unrepaired CPDs (after normalization) along the NTS of yeast genes is only slightly lower in the *elf1*-CTD Δ relative to WT (see Fig. 2f-g). We do not conclude from this slight difference that GG-NER of the NTS is necessarily faster in the *elf1*-CTD Δ mutant. However, if this slight difference in the fraction of unrepaired CPDs reflects a real increase in GG-NER activity, this could be potentially explained, for example, by changes in the nucleosome structure of transcribed genes due to the *elf1* mutant, since Elf1 facilitates transcription through nucleosomes, etc.

We respectfully disagree with Reviewer’s suggestion that normalizing using the alkaline gel analysis is somehow intrinsically problematic. The T4 endoV digestion and alkaline gel analysis measures the absolute number of CPDs at different repair time

points following UV irradiation, as previously described. We simply used these values to standardize the fraction of unrepaired CPDs in our CPD-seq data.

4. “Figure 3E is identical to Figure 1C. While recognizing the intention to compare the data of *elf1*-CTD Δ with *rad26* Δ here. However, presenting the same data in different figures does not appear to be necessary.”

RESPONSE: We removed Fig. 3e (and 3b) from the figure, as requested by the Reviewer.

5. “UV sensitivity and TC-NER data of *elf1* Δ , *elf1*-CTD Δ and *elf1*-Y99A are presented in different genetic backgrounds (WT, *rad16* Δ , *rad26* Δ and *rad16* Δ *rad26* Δ) across figures. This way of presentation makes it hard for readers to assess the degrees of TC-NER defects caused by different *elf1* mutants in different genetic backgrounds. The authors may consider presenting UV-sensitivity data in one multi-panel figure and repair data in another to enhance clarity for readers.”

RESPONSE: While we thank the Reviewer for this suggestion, we think that the current organization of the figures provides a more clear and logical summary of our findings.

Finally, we would like to thank each of the Reviewers for their helpful comments and suggestions. Their efforts have helped to significantly improve the manuscript.

Sincerely,

John Wyrick and Dong Wang
(on behalf of the co-authors)

References

1. Woudstra EC, *et al.* A Rad26-Def1 complex coordinates repair and RNA pol II proteolysis in response to DNA damage. *Nature* **415**, 929-933 (2002).
2. Damodaren N, Van Eeuwen T, Zamel J, Lin-Shiao E, Kalisman N, Murakami K. Def1 interacts with TFIIH and modulates RNA polymerase II transcription. *Proceedings of the National Academy of Sciences of the United States of America* **114**, 13230-13235 (2017).
3. Li S, Smerdon MJ. Rpb4 and Rpb9 mediate subpathways of transcription-coupled DNA repair in *Saccharomyces cerevisiae*. *The EMBO journal* **21**, 5921-5929 (2002).
4. Haradhvala NJ, *et al.* Mutational Strand Asymmetries in Cancer Genomes Reveal Mechanisms of DNA Damage and Repair. *Cell* **164**, 538-549 (2016).

5. Bohm KA, *et al.* Distinct roles for RSC and SWI/SNF chromatin remodelers in genomic excision repair. *Genome research* **31**, 1047-1059 (2021).
6. Selvam K, Plummer DA, Mao P, Wyrick JJ. Set2 histone methyltransferase regulates transcription coupled-nucleotide excision repair in yeast. *PLoS genetics* **18**, e1010085 (2022).
7. Prather D, Krogan NJ, Emili A, Greenblatt JF, Winston F. Identification and characterization of Elf1, a conserved transcription elongation factor in *Saccharomyces cerevisiae*. *Molecular and cellular biology* **25**, 10122-10135 (2005).
8. Mayer A, Lidschreiber M, Siebert M, Leike K, Söding J, Cramer P. Uniform transitions of the general RNA polymerase II transcription complex. *Nature structural & molecular biology* **17**, 1272-1278 (2010).
9. Okuda M, Nishimura Y. Extended string binding mode of the phosphorylated transactivation domain of tumor suppressor p53. *J Am Chem Soc* **136**, 14143-14152 (2014).
10. Di Lello P, *et al.* Structure of the Tfb1/p53 complex: Insights into the interaction between the p62/Tfb1 subunit of TFIIH and the activation domain of p53. *Molecular cell* **22**, 731-740 (2006).
11. Okuda M, Araki K, Ohtani K, Nishimura Y. The Interaction Mode of the Acidic Region of the Cell Cycle Transcription Factor DP1 with TFIIH. *Journal of molecular biology* **428**, 4993-5006 (2016).
12. Hodges AJ, Plummer DA, Wyrick JJ. NuA4 acetyltransferase is required for efficient nucleotide excision repair in yeast. *DNA repair* **73**, 91-98 (2019).
13. Geijer ME, *et al.* Elongation factor ELOF1 drives transcription-coupled repair and prevents genome instability. *Nature Cell Biology* **23**, 608-619 (2021).

REVIEWER COMMENTS

Reviewer #1 (Remarks to the Author):

The authors' responses answered most of my questions. Based on their new data and discussion, Elf1 plays an important but not indispensable role in both Rad26-dependent and -independent TCR through its interaction with p62. However, the last question of my first point, "how are Elf1 and Rad4 coordinated during TCR?", is not well addressed. Considering that Rad4 is required for TCR in the presence of Elf1, and the binding affinity of Rad4 to p62 is much higher than that of Elf1, I think there are two possible models of the role of Elf1 in TFIIH recruitment. First, there are parallel mechanisms to recruit TFIIH relying on Elf1, Rad4 and/or Def1, respectively. These pathways are independent of each other but they co-exist in yeast. Alternatively, these factors may collaborate to recruit TFIIH. That means, the role of Elf1 might be assisting Rad4 or Def1 to recruit TFIIH, so loss of Elf1 only reduce but not abolish TCR. However, the effects of Rad4 and Def1 on TCR are not shown in this manuscript. Notably, Rad4 and Def1 may also have other functions in TCR besides recruiting TFIIH. Anyway, the authors should discuss these possibilities and clarify the exact role of Elf1 in recruiting TFIIH as well as its relationship with Rad4 (or Def1).

I can understand the concerns of Reviewer #3, however, I think it is, to some degree, too hard on the authors.

It was shown in this manuscript that Elf1-CTD can interact with Tfb1-PH, and the lack of Elf1-CTD can compromise TCR and elevate UV sensitivity under certain genetic backgrounds. In my understanding, the major concern of Reviewer #3 is that the causation between the protein interaction and the phenotypes is not exclusive. I think the cause of this issue is the lack of a mutant that could completely abrogate Elf1-Tfb1 interaction without affecting any other process. The *elf1-CTDΔ* and *tfb1-PHΔ* mutants can completely abrogate their interaction, however, they may also affect other steps of TCR. In contrast, the *elf1-Y99A* mutant has minimal side-effect, but it can only partially inhibit Elf1-Tfb1 interaction, so the phenotypes were relatively weak or even undetectable under some conditions.

In addition, the yeast UV sensitivity assay for TCR genes should be performed in GGR-defective backgrounds, i.e., *rad16Δ* in this study. However, the effect of Elf1-CTD in TCR is weaker in *rad16Δ* mutant than in the WT strain. To explain this observation, the authors suggest in the Discussion (p18-19) that Rad4 might become more active in TCR when GGR is defective. Anyway, the effect of *elf1-CTDΔ* in *rad16Δ* background on UV sensitivity is weak. It became more obvious in *rad16Δrad26Δ* background, probably because Elf1-CTD (and Elf1-Y99) plays a more prominent (but still not indispensable) role in Rad26-independent TCR. Therefore, the UV sensitivity data are difficult for interpretation.

Here are my comments on Reviewer #3's points:

1. The manuscript asserts that the *elf1-Y99A* mutation is epistatic with the *tfb1-PHΔ* mutant. However, the *elf1-Y99A* mutant does not exhibit UV sensitivity across various genetic backgrounds, including *rad16Δ*, *rad16Δ rad26Δ*, and *rad16Δ rad26Δ tfb1-PHΔ* (Supplementary Figure 12). This observation implies that the *elf1-Y99A* mutation does not notably influence TC-NER, even in the absence of RAD26.

Since Elf1-Y99 is not required for Rad26-dependent TCR (Supplementary Figure 13), it is not surprising that *elf1-Y99A* could not increase UV sensitivity in *rad16Δ* background. Indeed, there seemed to be a weak effect in Supplementary Figure 10, whereas no effect in Supplementary Figure 12a, showing a subtle difference. For the *rad16Δrad26Δtfb1-PHΔ* background, the authors' point is that Elf1-Y99A should not increase the UV sensitivity in *tfb1-PHΔ* background. Regarding to *rad16Δrad26Δ* background, I think there is obvious difference between *rad16Δrad26Δ* and *rad16Δrad26Δelf1-Y99A* in Supplementary Figure

12b (comparing the first and second rows). It was also shown in Figure 5b and quantified in Figure 5c. Moreover, Figure 5g-i showed that Elf1-Y99 has a weak effect on Rad26-independent TCR, although this effect is weaker than Elf1-CTD. I think these results suggest that Elf1-Y99 does play a role in Rad26-independent TCR, although the effect is not strong. It is not conflict with hypothesis that Elf1-CTD contributes to TCR by recruiting TFIIH since the Y99A mutant only weakens the interaction between Elf1-CTD and Tfb1-PH by about 50% (Figure 5d-f).

2. Furthermore, the claim that "the *elf1-CTDΔ* mutant demonstrates epistasis with *tfb1-PHΔ*, particularly in the *rad16Δ rad26Δ* mutant background" does not seem to hold upon closer inspection. While the *elf1-CTDΔ* mutation mildly heightened UV sensitivity in *rad16Δ rad26Δ* cells, it did not notably increase UV sensitivity in *rad16Δ* cells alone (as shown in Figure 2b).

As discussed above, the effects of *elf1-CTDΔ* in *rad16Δ* are minor. However, I think *elf1-CTDΔrad16Δ* is slightly more sensitive than *rad16Δ* in Figure 2b (comparing 10 J/m² and 25 J/m²). And the effect of *elf1-CTDΔ* in *rad16Δrad26Δ* is very strong rather than mild.

However, cells with *rad16Δ elf1-CTDΔ tfb1-PHΔ* mutations exhibited greater UV sensitivity compared to *rad16Δ tfb1-PHΔ* cells (Supplementary Figure 12a), indicating a synergistic effect rather than epistasis. This interaction suggests that the role of Elf1-CTD in TC-NER is independent of Tfb1-PH.

Although the *rad16Δelf1-CTDΔtfb1-PHΔ* strain is slightly more sensitive than *rad16Δtfb1-PHΔ* strain, I think the difference is comparable with that between *rad16Δelf1-CTDΔ* and *rad16Δ*, which is not notable due to Reviewer #3 (see above). In contrast, *rad16Δelf1-CTDΔtfb1-PHΔ* is much more sensitive than *rad16Δelf1-CTDΔ*.

Moreover, the *rad16Δ rad26Δ elf1-CTDΔ tfb1-PHΔ* combination showed a UV sensitivity at least five times greater than that of *rad16Δ tfb1-PHΔ* (Supplementary Figure 12b), challenging the assertion of epistasis between *elf1-CTDΔ* and *tfb1-PHΔ*.

I don't know why the *rad16Δrad26Δelf1-CTDΔtfb1-PHΔ* need to be compared with *rad16Δtfb1-PHΔ*, since the role of Rad26 is not the scope of this study. If the reviewer means *rad16Δrad26Δtfb1-PHΔ* rather than *rad16Δtfb1-PHΔ*, I think the difference is just minor, comparable with those marked with red boxes in above figures.

Taken together, *elf1-CTDΔ* does have an additional effect in *tfb1-PHΔ* background, implying the roles of Elf1-CTD besides interacting with Tfb1-PH. However, the additional effect is minor, thus it is still reasonable to claim that “The *elf1-CTDΔ* mutant was also largely epistatic with *tfb1-PHΔ*, particularly in the *rad16Δ rad26Δ* mutant background.” (This is the author's exact words, and they just claimed “largely” epistatic.) Anyway, I will suggest the authors to quantify the UV sensitivity results (as Figure 5c) to avoid debates on whether there are (strong) differences between different strains.

3. The manuscript fails to present data evaluating the direct impact of *tfb1-PHΔ* on TC-NER or to compare the TC-NER deficiency in cells with combined mutations (*tfb1-PHΔ* and *elf1-CTDΔ*) against those with single mutations.

This is a good suggestion. If it is technically available, the authors should measure TCR by CPD-seq in those strains rather than just comparing UV sensitivity. As discussed above, UV sensitivity should be assessed in *rad16Δ* background, which may weaken the impact of Elf1-CTD on TCR.

In summary, I think it is reasonable to claim that the interaction between Elf1-CTD and Tfb1-PH contributes to TCR, although there are still doubts about the importance of this contribution. In addition to my original suggestion for the authors to clarify the role of Elf1 in recruiting TFIIH, I will ask them to quantify the UV sensitivity results and directly measure TCR in *tfb1-PHΔ*-related stains, including *tfb1-PHΔ*, *tfb1-PHΔelf1-CTDΔ* and *tfb1-PHΔelf1-Y99A*. These data can help clarify the importance of Elf1-Tfb1 interaction in TCR.

Reviewer #2 (Remarks to the Author):

I am satisfied with the authors' responses to my comments. I think this is an interesting paper that would be perfectly appropriate for Nature Communications.

Reviewer #3 (Remarks to the Author):

The authors have addressed certain concerns I raised; however, the revised manuscript still lacks adequate evidence to substantiate the key conclusion that Elf1-CTD's involvement in TC-NER operates through Tfb1-PH.

1. The manuscript asserts that the *elf1-Y99A* mutation is epistatic with the *tfb1-PHΔ* mutant. However, the *elf1-Y99A* mutant does not exhibit UV sensitivity across various genetic backgrounds, including *rad16Δ*, *rad16Δ rad26Δ*, and *rad16Δ rad26Δ tfb1-PHΔ* (Supplementary Figure 12). This observation implies that the *elf1-Y99A* mutation does not notably influence TC-NER, even in the absence of RAD26.

2. Furthermore, the claim that "the *elf1-CTDΔ* mutant demonstrates epistasis with *tfb1-PHΔ*, particularly in the *rad16Δ rad26Δ* mutant background" does not seem to hold upon closer inspection. While the *elf1-CTDΔ* mutation mildly heightened UV sensitivity in *rad16Δ rad26Δ* cells, it did not notably increase UV sensitivity in *rad16Δ* cells alone (as shown in Figure 2b). However, cells with *rad16Δ elf1-CTDΔ tfb1-PHΔ* mutations exhibited greater UV sensitivity compared to *rad16Δ tfb1-PHΔ* cells (Supplementary Figure 12a), indicating a synergistic effect rather than epistasis. This interaction suggests that the role of Elf1-CTD in TC-NER is independent of Tfb1-PH. Moreover, the *rad16Δ rad26Δ elf1-CTDΔ tfb1-PHΔ* combination showed a UV sensitivity at least five times greater than that of *rad16Δ tfb1-PHΔ* (Supplementary Figure 12b), challenging the assertion of epistasis between *elf1-CTDΔ* and *tfb1-PHΔ*.

3. The manuscript fails to present data evaluating the direct impact of *tfb1-PHΔ* on TC-NER or to compare the TC-NER deficiency in cells with combined mutations (*tfb1-PHΔ* and *elf1-CTDΔ*) against those with single mutations.

Response to Reviewers

Reviewer #1

1. "The authors' responses answered most of my questions. Based on their new data and discussion, Elf1 plays an important but not indispensable role in both Rad26-dependent and -independent TCR through its interaction with p62. However, the last question of my first point, "how are Elf1 and Rad4 coordinated during TCR?", is not well addressed. Considering that Rad4 is required for TCR in the presence of Elf1, and the binding affinity of Rad4 to p62 is much higher than that of Elf1, I think there are two possible models of the role of Elf1 in TFIIH recruitment. First, there are parallel mechanisms to recruit TFIIH relying on Elf1, Rad4 and/or Def1, respectively. These pathways are independent of each other but they co-exist in yeast. Alternatively, these factors may collaborate to recruit TFIIH. That means, the role of Elf1 might be assisting Rad4 or Def1 to recruit TFIIH, so loss of Elf1 only reduce but not abolish TCR. However, the effects of Rad4 and Def1 on TCR are not shown in this manuscript. Notably, Rad4 and Def1 may also have other functions in TCR besides recruiting TFIIH. Anyway, the authors should discuss these possibilities and clarify the exact role of Elf1 in recruiting TFIIH as well as its relationship with Rad4 (or Def1)."

RESPONSE: We thank the Reviewer for this interesting suggestion. A recently published study indicates that the Rad4 TFIIH-interacting region (TIR) plays a role in TCR (and GGR) by binding to the PH domain of p62¹. We favor the first model mentioned by the Reviewer, namely that there are parallel and potentially redundant mechanisms by which TFIIH is recruited by Elf1 or Rad4 TIR. Consistent with this model, we now include new data indicating that the *elf1*-CTD Δ mutant shows enhanced UV sensitivity when mutated in conjunction with the *rad4*-TIR Δ in both *rad16* Δ and *rad16* Δ *rad26* Δ mutant backgrounds (see Supplementary Fig. 14). We describe these new data and the corresponding model that these data suggest on page 19 of the revised manuscript.

Reviewer #2

"I am satisfied with the authors' responses to my comments. I think this is an interesting paper that would be perfectly appropriate for Nature Communications."

RESPONSE: We thank the Reviewer for their comments.

Reviewer #3

0. "The authors have addressed certain concerns I raised; however, the revised manuscript still lacks adequate evidence to substantiate the key conclusion that Elf1-CTD's involvement in TC-NER operates through Tfb1-PH."

RESPONSE: Our data indicate that the Elf1-CTD plays a role in TC-NER in yeast and binds the Tfb1-PH domain *in vitro*. Furthermore, we show that specific mutations in the Elf1 TFIIH-interacting region (TIR) that we identified (i.e., Elf1-Y99A) affect Tfb1-PH domain binding *in vitro* and cause enhanced UV sensitivity and a TC-NER defect in *rad26* Δ mutant cells (Fig. 5 and new Supplementary Fig. 12c). These data are

consistent with the model that the TIR in the Elf1 CTD regulates TC-NER by binding the Tfb1 PH domain. We now include new quantitative UV sensitivity data that clearly indicate that the *elf1*-Y99A mutant is epistatic with the *tfb1*-PH Δ in a *rad16* Δ *rad26* Δ mutant background (Supplementary Fig. 12c). These data confirm our model that the TIR in the Elf1 CTD regulates TC-NER through a functional interaction with the Tfb1 PH domain.

1. "The manuscript asserts that the *elf1*-Y99A mutation is epistatic with the *tfb1*-PH Δ mutant. However, the *elf1*-Y99A mutant does not exhibit UV sensitivity across various genetic backgrounds, including *rad16* Δ , *rad16* Δ *rad26* Δ , and *rad16* Δ *rad26* Δ *tfb1*-PH Δ (Supplementary Figure 12). This observation implies that the *elf1*-Y99A mutation does not notably influence TC-NER, even in the absence of RAD26."

RESPONSE: The Reviewer's comment here is a bit perplexing, given the data presented in the manuscript. The quantitative UV sensitivity data shown in Fig. 5c clearly indicates that the *elf1*-Y99A mutant significantly increases UV sensitivity in a *rad16* Δ *rad26* Δ mutant background for both the 10 and 15 J/m² UV doses ($P < 0.05$). We have since performed new quantitative UV sensitivity data for the *elf1*-Y99A mutant in a *rad16* Δ *rad26* Δ mutant background (Supplementary Fig. 12c), which yielded similar results. Across both of these independent sets of experiments, the *elf1*-Y99A mutant increases UV sensitivity an average of ~5-fold for the 10 and 15 J/m² UV doses. Importantly, we observed no significant difference in UV sensitivity in the *elf1*-Y99A *tfb1*-PH Δ mutant relative to *tfb1*-PH Δ alone in the *rad16* Δ *rad26* Δ background (Supplementary Fig. S12c), indicating that the *elf1*-Y99A mutant is epistatic with a deletion in the *TFB1*-PH domain. These quantitative data are consistent with our conclusions from the UV spotting data shown in Supplementary Fig. 12b. Taken together, these findings demonstrate that the *elf1*-Y99A mutant does cause significantly UV sensitivity in a *rad16* Δ *rad26* Δ mutant background, and this UV sensitivity is epistatic with a deletion in the *TFB1*-PH domain, confirming that they function in the same TC-NER pathway in yeast. These new quantitative data are discussed on pages 14 and 15 of the revised manuscript.

While we agree that the *elf1*-Y99A mutant alone exhibited little to no effect on UV sensitivity in a *rad16* Δ mutant background, the *elf1*-Y99A mutant clearly enhances the UV sensitivity of the *rad26* Δ mutant in a *rad16* Δ background (see Fig. 5c and Supplementary Fig. 12c). Like *rad26* Δ , the *tfb1*-PH Δ mutant affects TC-NER¹ and causes UV sensitivity in a *rad16* Δ mutant background (Supplemental Fig. 12a). However, the *tfb1*-PH Δ mutant does not show elevated UV sensitivity in conjunction with *elf1*-Y99A (Supplemental Fig. 12a), unlike *rad26* Δ . Taken together, these data support the hypothesis that Elf1-Y99 is epistatic with the *TFB1*-PH domain, but is not epistatic with Rad26.

2. 'Furthermore, the claim that "the *elf1*-CTD Δ mutant demonstrates epistasis with *tfb1*-PH Δ , particularly in the *rad16* Δ *rad26* Δ mutant background" does not seem to hold upon closer inspection. While the *elf1*-CTD Δ mutation mildly heightened UV sensitivity in *rad16* Δ *rad26* Δ cells, it did not notably increase UV sensitivity in *rad16* Δ cells alone (as shown in Figure 2b). However, cells with *rad16* Δ *elf1*-CTD Δ *tfb1*-PH Δ mutations

exhibited greater UV sensitivity compared to *rad16Δ tfb1-PHΔ* cells (Supplementary Figure 12a), indicating a synergistic effect rather than epistasis. This interaction suggests that the role of Elf1-CTD in TC-NER is independent of Tfb1-PH. Moreover, the *rad16Δ rad26Δ elf1-CTDΔ tfb1-PHΔ* combination showed a UV sensitivity at least five times greater than that of *rad16Δ tfb1-PHΔ* (Supplementary Figure 12b), challenging the assertion of epistasis between *elf1-CTDΔ* and *tfb1-PHΔ*.

RESPONSE: Our analysis of the spotting data shown in Supplementary Fig. 12 led us to conclude that the *elf1-CTDΔ* was “largely epistatic with *tfb1-PHΔ*, particularly in the *rad16Δ rad26Δ* mutant background” (see page 15 of manuscript). We used this terminology (i.e., ‘largely epistatic’) because our spotting data suggested that the UV sensitivity of the *elf1-CTDΔ tfb1-PHΔ* was slightly more sensitive than the *tfb1-PHΔ* mutant alone in both the *rad16Δ* (Supplementary Fig. 12a) and *rad16Δ rad26Δ* (Supplementary Fig. 12b) backgrounds. Reviewer 1 came to essentially the same conclusion after reanalyzing these same data.

In the revised manuscript, we now include quantitative UV sensitivity data addressing this question (see Supplementary Fig. 12d). The *elf1-CTDΔ* results in an average ~23-fold increase in UV sensitivity across the various UV doses in a *rad16Δ rad26Δ* background, consistent with the UV sensitivity reported in our original quantitative UV sensitivity data (average ~25-fold more UV sensitive in Fig. 5c). In the *elf1-CTDΔ tfb1-PHΔ rad16Δ rad26Δ* mutant, this UV sensitivity increases an additional ~2.6-fold relative to the *tfb1-PHΔ rad16Δ rad26Δ* mutant alone, and this increase is statistically significant at the 7.5 and 10 J/m² UV doses (P < 0.05, Supplementary Fig. 12d). These findings are consistent with the original spotting data indicating that the *rad16Δ rad26Δ elf1-CTDΔ tfb1-PHΔ* is slightly more sensitive than the *rad16Δ rad26Δ tfb1-PHΔ* mutant (Supplementary Fig. 12b).

In summary, our quantitative data indicate that the *elf1-CTDΔ* mutant has a much larger effect on UV sensitivity in *rad16Δ rad26Δ* cells in which the TFB1-PH domain is intact (~23-fold increase in UV sensitivity) rather than when the TFB1-PH domain is deleted (~2.6-fold increase in UV sensitivity), consistent with the conclusion that the Elf1-CTD is largely epistatic with the TFB1-PH domain. The somewhat elevated UV sensitivity of the *tfb1-PHΔ elf1-CTDΔ* double mutant in the *rad16Δ* and *rad16Δ rad26Δ* mutant backgrounds likely signifies that the Elf1 CTD can regulate TC-NER not only by binding the TFIIH PH domain, but potentially by a second, PH domain-independent mechanism. Consistent with this hypothesis, a recent study indicates that there is residual TC-NER activity in yeast lacking the Tfb1-PH domain¹. We hope to investigate this alternative mechanism by which the Elf1-CTD regulates repair in future studies. These new findings are discussed on pages 15 of the revised manuscript.

The Reviewer’s observation that the “*rad16Δ rad26Δ elf1-CTDΔ tfb1-PHΔ* combination showed a UV sensitivity at least five times greater than that of *rad16Δ tfb1-PHΔ* (Supplementary Figure 12b)” does not necessarily challenge “the assertion of epistasis between *elf1-CTDΔ* and *tfb1-PHΔ*,” as stated by the Reviewer. Instead, it primarily indicates that the *rad26Δ* mutant is not epistatic with the *tfb1-PHΔ* mutant, as can be ascertained by comparing the UV sensitivity (at 15 J/m²) of the *rad16Δ tfb1-PHΔ* and *rad16Δ rad26Δ* mutants to that of the *rad16Δ rad26Δ tfb1-PHΔ* mutant (Supplementary Fig. 12a,b). This is a potentially interesting observation, as it suggests

that Rad26 promotes TC-NER largely independent of the TFB1-PH domain, but is not the focus of the current study.

3. “The manuscript fails to present data evaluating the direct impact of *tfb1-PHΔ* on TC-NER or to compare the TC-NER deficiency in cells with combined mutations (*tfb1-PHΔ* and *elf1-CTDΔ*) against those with single mutations.”

RESPONSE: A new paper published by Gong et al. ¹ indicates that the Tfb1-PH domain regulates TC-NER in yeast. We now mention this result on page 14, paragraph 2 of the revised manuscript. Since this result is now published, we have not pursued studies characterizing the impact of the Tfb1-PH domain on TC-NER in the current manuscript.

Gong et al. also demonstrated that the Tfb1-PH domain is required for efficient GG-NER¹. This dual role precludes us from directly analyzing the effects on repair of double mutants of the *tfb1-PHΔ* and *elf1-Y99A* (or *CTDΔ*) mutants, since the added effect on GG-NER activity would significantly complicate the resulting repair patterns and subsequent analysis. While it is possible to analyze these mutant combinations in a GG-NER defective strain (i.e., *rad16Δ*), in this study we have primarily focused our analysis on analyzing repair in GG-NER competent yeast. Thus, these new experiments would require significant time and expense, and therefore we view it as beyond the scope of the current study. Moreover, the UV sensitivity data shown in Supplementary Fig. 12 clearly indicate that the Elf-Y99 is epistatic with the TFB1-PH domain. For these reasons, we have not performed these experiments in the current manuscript, but hope to do so in a future study.

Sincerely,

John Wyrick and Dong Wang
(on behalf of the co-authors)

References

1. Gong W, Holmberg H, Lu C, Huang M, Li S. Interplay of the Tfb1 pleckstrin homology domain with Rad2 and Rad4 in transcription coupled and global genomic nucleotide excision repair. *Nucleic acids research*, (2024).